# LEARNING TO EVICT FROM KEY-VALUE CACHE

## ABSTRACT

The growing size of Large Language Models (LLMs) makes efficient inference challenging, primarily due to the memory demands of the autoregressive Key-Value (KV) cache. Existing eviction or compression methods reduce cost but rely on heuristics, such as recency or past attention scores, which serve only as indirect proxies for a token's future utility and introduce computational overhead. We reframe KV cache eviction as a reinforcement learning (RL) problem: learning to rank tokens by their predicted usefulness for future decoding. To this end, we introduce KV Policy (KVP), a framework of lightweight per-head RL agents trained on pre-computed generation traces using only key and value vectors. Each agent learns a specialized eviction policy guided by a holistic reward, derived from future utility, that evaluates the quality of the ranking across all cache budgets, requiring no modifications to the underlying LLM or additional inference. Evaluated on the long-context benchmark RULER and the multi-turn dialogue benchmark OASST2-4k, KVP significantly outperforms baselines. Furthermore, zero-shot tests on standard downstream tasks indicate that KVP generalizes well beyond its training distribution. These results demonstrate that learning to predict future token utility is a powerful and scalable paradigm for adaptive KV cache management.

## 1 INTRODUCTION

Large Language Models (LLMs) based on the Transformer architecture (Vaswani et al., 2017) have revolutionized natural language processing, demonstrating remarkable capabilities across a wide range of tasks (Brown et al., 2020; Önden & Alnour, 2023; Touvron et al., 2023). However, their practical deployment, especially for applications involving long sequences or interactive sessions, is often burdened by substantial computational requirements during inference. A critical bottleneck arises from the Key-Value (KV) cache, a mechanism inherent to autoregressive generation in Transformers (Sheng et al., 2023). This cache stores the keys and values of previous tokens, avoiding redundant computations but growing linearly with the input and generated sequence length. For long contexts, the KV cache can consume tens or even hundreds of gigabytes of memory, rapidly exceeding the capacity of modern hardware accelerators and necessitating strategies for efficient management.

To address this memory challenge, a variety of KV cache management techniques have been proposed. These range from simple heuristics like keeping only the most recent tokens (sliding window), to more sophisticated methods that leverage insights into attention patterns. Approaches like H2O, SnapKV, and TOVA Zhang et al. (2023); Li et al. (2024); Oren et al. (2024) use signals such as past attention scores or attention sinks to identify and retain important tokens. Others employ quantization to reduce the memory footprint (Dettmers et al., 2022; Xiao et al., 2023) or use low-rank approximations to represent the cache state more compactly (Singhania et al., 2024; Ribar et al., 2024).

While these hand-crafted policies have advanced the state of the art, they are fundamentally built on heuristics that serve as indirect proxies for a token's future importance. They assume that what was important before will remain important. This assumption might not always hold, as the informational needs of generation are dynamic and content-dependent. A token's true value is determined by its utility to future decoding steps, a quantity these methods do not directly optimize for. Consequently, they are prone to suboptimal eviction decisions, leading to an irrecoverable loss of critical information and degraded generation quality. Even post-hoc compression methods based on attention signals are fundamentally backward-looking and have additional drawbacks: they require repeated computation of attention statistics, which introduces significant overhead; and they are not compatible with efficient implementations like FlashAttention (Dao et al., 2022), limiting their practicality.

| KVP | Future Attention Importance | StreamingLLM |
|---|---|---|
| I'm planning a trip to Rome next month. I want to visit the Colosseum, the Vatican Museums, and the Trevi Fountain. What's the best order to visit these attractions to minimize travel time? <\|im_end\|>Also, should I book tickets in advance for these places?<\|im_end\|> | I'm planning a trip to Rome next month. I want to visit the Colosseum, the Vatican Museums, and the Trevi Fountain. What's the best order to visit these attractions to minimize travel time? <\|im_end\|>Also, should I book tickets in advance for these places?<\|im_end\|> | I'm planning a trip to Rome next month. I want to visit the Colosseum, the Vatican Museums, and the Trevi Fountain. What's the best order to visit these attractions to minimize travel time? <\|im_end\|>Also, should I book tickets in advance for these places?<\|im_end\|> |

Figure 1: **Future token importance for KV cache eviction.** Effective KV cache eviction requires identifying tokens that will receive little or no future attention. *(center)* We roll out a sample KV cache and measure the true cumulative future attention for each token, then rank tokens by this importance and color them accordingly (bright = high rank, white = low). *(right)* Importance estimated by the fixed sink-and-recency heuristic of *StreamingLLM* deviates substantially from true importance ranking. *(left)* Our learned policy closely recovers the complex, non-local structure of future attention despite using only past keys and values, without access to queries, attention scores, or future tokens. *We expand the comparison with all strategies in Figure 7.*

In this work, we move beyond hand-crafted policies and reframe KV cache eviction as a Reinforcement-ment Learning (RL) problem: learning an attention-free policy that selects which tokens to evict. We formulate this learning problem as a ranking task, where the agent orders tokens by their predicted future utility. As qualitatively illustrated in Figure 1, a learned policy can successfully approximate a token's future utility, a complex, non-local structure, where simple heuristics fail. Such a ranking enables a highly efficient and flexible strategy: eviction is performed by discarding the lowest-ranked entries to meet any memory budget. As a reward signal, we define the ranking successful if for any given cache size, the most valuable information is retained, thus minimizing performance degradation.

We introduce KV Policy (KVP), a framework that trains a distinct, lightweight RL agent for each KV head in the model. This per-head specialization allows each policy to adapt to the unique attentional patterns of its corresponding head. The agents are trained efficiently on pre-computed generation traces without any additional inference, using only the key and value vectors as input and requiring no architectural changes to the underlying LLM. To train the agents, we introduce a reward that holistically evaluates the quality of the policy's sorting for all possible cache budgets. For every possible cache budget, we measure how much essential information in the future would be erroneously evicted with the candidate sorting when only the top tokens from the cache are kept.

Evaluated on long-context synthetic benchmark RULER (Hsieh et al., 2024) and a multi-turn natural language benchmark OASST2-4k (Köpf et al., 2023), KVP significantly outperforms strong heuristic baselines, demonstrating that learning specialized policies to predict the future utility of tokens is a powerful and scalable paradigm for KV cache eviction. Evaluation in a zero-shot generalization setting on standard downstream benchmarks from the EleutherAI Language Model Evaluation Harness (Gao et al., 2024) suggests that KVP retains strong performance even out of distribution.

In summary, our main contributions are:

- We reframe KV cache eviction as a learning problem: ranking cache entries by their predicted future utility.
- We introduce KVP, a system of lightweight, per-head RL agents that learn specialized sorting policies using only key and value vectors without using any attention information.
- We propose a holistic reward that evaluates eviction policies across all cache budgets without additional LLM inference.
- We show that KVP substantially improves long-context performance over strong baselines and generalizes to unseen domains.

## 2 RELATED WORK

**Attention-Based Eviction.** Eviction-based approaches aim to choose a subset of the KV cache. Early methods used simple heuristics like First-In-First-Out (FIFO) or Least Recently Used (LRU) (Xiao et al., 2024). More recent work leverages the inherent sparsity in attention patterns. Techniques like StreamingLLM (Xiao et al., 2024) and H2O (Zhang et al., 2023) observe that initial tokens ("attention sinks") and recent tokens often capture most of the required context, allowing for the eviction of intermediate tokens. Others, such as KeyFormer (Adnan et al., 2024) and related works

(Cai et al., 2024), explicitly analyze attention scores or structures to identify and retain important tokens, sometimes using the current query to inform eviction (Li et al., 2024; Lee et al., 2024a). Although our work is an eviction methodology, it departs from this paradigm by instead learning a *forward-looking, query-independent policy* to directly predict a token's future utility, rather than inferring it from past attention or the current query.

**Memory Hierarchy Management.** Recognizing the limited size of fast GPU memory, some approaches utilize system memory (CPU RAM) as a secondary cache layer (Chen et al., 2024b; Sheng et al., 2023). Recent sparse retrieval approaches, such as IceCache (Anonymous, 2025), ArkVale (Chen et al., 2024a), MagicPig (Chen et al., 2025), and InfiniGen (Lee et al., 2024b), manage these hierarchies by offloading KV entries to slower memory and retrieving them only when needed, possibly in pages (Tang et al., 2024). While this allows for effectively larger cache sizes, it introduces significant latency (a reload cost) when accessing offloaded entries. Policies for deciding what and when to offload are often heuristic. While our work focuses on eviction, the learned ranking it produces provides a principled, data-driven signal for such hierarchical management: the lowest-ranked entries are natural candidates for being moved to slower memory, representing a powerful synergy between our approaches.

**Representation Compression.** Instead of removing entries, another line of work focuses on reducing the memory required per entry. Quantization techniques reduce the numerical precision (e.g., to 8-bit integers or lower) of keys and values, significantly cutting memory usage, often with minimal performance impact (Dettmers et al., 2022; Xiao et al., 2023), while low-rank approximation methods represent the key and value matrices using lower-dimensional projections (Singhania et al., 2024; Ribar et al., 2024). A distinct approach is state merging, where methods like MorphKV (Ghadia et al., 2025) identify and combine semantically similar KV pairs into new, synthetic representations. These techniques are orthogonal and complementary to eviction strategies, as one could utilize our approach to filter out the least useful tokens and then apply MorphKV's merging logic to the remaining high-utility tokens.

**Learned Approaches.** Applying machine learning to directly optimize cache management policies is less common than heuristic approaches. While learning has been used extensively for general caching problems (Afrin et al., 2024; Shuja et al., 2020; Wang & Friderikos, 2020), its application specifically to the dynamic nature of the Transformer KV cache is emerging (Chari et al., 2025; Cetin et al., 2024; Nawrot et al., 2024; Ge et al., 2024). Some recent works, such as Gisting Token (Deng et al., 2025) and Activation Beacon (Zhang et al., 2025), learn to compress context into compact summary tokens or condensed activations. Our work differs by learning a fine-grained ranking over all tokens rather than a summary; however, these approaches are complementary, as KVP could identify which tokens are best suited for summarization. Our approach is distinguished by utilizing RL to train a lightweight, per-head policy. A novel reward signal, the eviction error across all cache budgets, ensures robust performance under varying memory limits. Furthermore, the per-head rankings and future attention estimates from KVP could directly inform a dynamic, non-uniform budget allocation, a promising direction for future work.

Overall, while prior work has addressed KV cache constraints through heuristics, memory hierarchies, or compression, our approach introduces an RL framework that casts eviction as a ranking problem. By training lightweight, per-head policies to predict tokens' future utility and optimizing them with a global, budget-agnostic reward, we offer an adaptive, query-independent solution. Crucially, this method is also complementary to many existing techniques, and may provide a principled signal for memory offloading, context gisting, or dynamic budget allocation.

## 3 METHODOLOGY: LEARNING TO EVICT KV CACHE ENTRIES

We consider the problem of KV cache eviction: given $n$ tokens $\mathcal{X} = \{x_i\}_{i \in [n]}$ and a budget $b$, select a subset $\mathcal{S}^\star \subset \mathcal{X}$ maximizing some downstream performance reward $R$ as

$$\mathcal{S}_b^\star \in \arg\max_{|\mathcal{S}|=b, \mathcal{S} \subset \mathcal{X}} R(\mathcal{S}). \tag{1}$$

Because generation is autoregressive and budgets vary over time, a practical solution must handle arbitrary $n$ and all $b \in [n]$. Although this selection problem is NP–hard even for linear rewards (Nemhauser et al., 1978), it becomes structured under two mild conditions: **(i) uniqueness:**

each $\mathcal{S}_b^\star$ is unique and **(ii) nestedness:** $\mathcal{S}_b^\star \subset \mathcal{S}_{b+1}^\star$ for all $b$. Uniqueness can be straightforwardly achieved with simple tie-breaking and nestedness is natural because tokens essential under a small budget should remain so as capacity grows. Under these constraints KV cache eviction problem is equivalent to KV cache ranking *with proof deferred to Appendix A.1.*

**Proposition 1.** *Assume (i) **uniqueness** of each $S_b^\star$ and (ii) **nestedness**: $S_b^\star \subset S_{b+1}^\star$ for all $b$. Then, there exists a total order (ranking) $\pi$ such that*

$$S_b^\star = \{\, i \in [n] : \ \pi(i) \le b \,\} \quad \text{for all } b.$$

*Equivalently, there exists a scoring function whose top-$b$ elements realize $S_b^\star$ for every $b$.*

Proposition 1 lets us reformulate eviction as learning a single budget-agnostic scoring function. At any generation step, the cache entries are ordered from most to least valuable for future decoding using the learned scoring function; for a given budget $b$, we retain the top-$b$. A high-quality ranking preserves critical information across all budgets, minimizing degradation.

To learn this scoring function, we adopt a reinforcement-learning (RL) approach. We parameterize a stochastic ranking policy and directly optimize the true discrete end-to-end reward using policy-gradient methods. We later analyze this choice against differentiable relaxations of sorting (Prillo & Eisenschlos, 2020; Grover et al., 2019; Blondel et al., 2020)) in Section 4.3.

We employ a lightweight RL agent, governed by parameters $\theta$, to define the scoring function $f(;\theta)$. To capture the specialized functions of different attention mechanisms, we train a distinct agent for each KV head in the LLM. To this end, we introduce KVP, a framework for efficiently training lightweight, per-head RL agents to perform this ranking. In the remaining of this section, we first discuss the agent architecture, then the reward, and finally the learning process.

## 3.1 KV Cache Eviction Agent

In order to formulate learning to sort as an RL problem, we largely follow the Plackett-Luce model (Plackett, 1975). Given a set of input tokens $\{x_i\}_{i \in [N]}$, we learn a parametric scoring function $f(x_i; \theta)$ which assigns a score to each input token $x_i$. This scoring function induces a stochastic sorting policy $\pi_\theta$ which samples permutation $\sigma = (\sigma_1, \ldots, \sigma_N)$ sequentially as;

$$\pi_\theta(\sigma | x_1, \ldots, x_N) = \prod_{i=1}^{N} \frac{\exp\left(f(x_{\sigma_i}; \theta)\right)}{\sum_{j=i}^{N} \exp\left(f(x_{\sigma_j}; \theta)\right)} \tag{2}$$

At each step $i$, the next element $\sigma_i$ is sampled proportionally to its score, normalized over the remaining tokens. This process defines a valid distribution over all permutations. The scoring function $f(;\theta)$ together with the sampling policy, forms our KV cache eviction agent. We next specify the parameterization of $f(;\theta)$.

**Scoring Function** The representation for token $x_i$ is the concatenation of its key vector $k_i$, value vector $v_i$, and its original position $pos_i$ as $x_i = (k_i, v_i, pos_i)$. We define the scoring function $f(;\theta)$ as a small Multi-Layer Perceptron (MLP) parametrized with $\theta$ as $f(x_i\theta) = MLP_\theta(k_i, v_i, pos_i)$. Crucially, the policy relies only on information available in the cache and does not require access to future information, previous attention scores or any query embedding.

**Efficient Parallel Sampling with Gumbel-Sort** While Equation (2) defines the distribution, sequential sampling is inefficient. Fortunately, a permutation can be sampled from this distribution in a single step using the Gumbel-Sort (Mena et al., 2018). Given the scores $f(x_i; \theta)$, we generate i.i.d. noise samples $g_i \sim \text{Gumbel}(0, 1)$. A permutation $\sigma$ is then sampled by sorting the perturbed scores:

$$\sigma = \texttt{argsort}_{i \in [N]} \left(f(x_i; \theta) + g_i\right) \tag{3}$$

This procedure is non-autoregressive, fully parallelizable on modern hardware, and allows us to sample an entire permutation with just one forward pass of the scoring model and a fast sort operation. This is critical for efficient training.

### 3.1.1 GLOBAL REWARD FOR OFFLINE RL

A key component of our framework is a reward signal that globally evaluates the quality of an agent's entire ranked output, directly optimizing for the efficient preservation of information across all possible cache budgets. More importantly we define this reward in a way to enable training without any additional LLM inference.

Consider the input set of $n$ tokens $\mathcal{X} = \{x_i\}_{i \in [n]}$ and a candidate permutation $\{\sigma_i\}_{i \in [n]}$, ordered from most to least important. The total reward is defined as the sum of reward over all possible target cache sizes $1 \le b \le n-1$. Since the kept tokens are always the top-$b$, this translates into

$$\mathcal{R}(\sigma_1, \dots, \sigma_n; \mathcal{X}) = \sum_{b=1}^{n-1} \mathcal{R}^b(\sigma_{n-b}, \dots, \sigma_n; \mathcal{X}) \tag{4}$$

where $\mathcal{R}^b$ is the reward for a specific budget $b$.

We define $\mathcal{R}^b$ based on an importance score derived from the future attention patterns. Consider $f$ future tokens ($x_{n+1}$ to $x_{n+f}$) in the training data after $n$ input tokens and the attention as $A(x_i, x_j)$, we define the importance of token $x_i$ as the total future attention it accumulates: $\sum_{j=n+1}^{n+f} A(x_i, x_j)$.

Given the permutation $\sigma$, a cache of budget $b$ retains the top-$b$ tokens $\sigma_1, \dots, \sigma_b$ and evicts the rest. The cost for this budget is the total importance of the evicted tokens. We define the per-budget reward $\mathcal{R}^b$ as the negative of this cost for all evicted tokens:

$$\mathcal{R}^b(\sigma_1, \dots, \sigma_n; \mathcal{X}) = - \sum_{i=b+1}^{n} \sum_{j=n+1}^{n+f} A(x_{\sigma_i}, x_j) \tag{5}$$

For models with Grouped-Query Attention (Ainslie et al., 2023), the attention score $A(x_i, x_j)$ from a future query group is the maximum attention value across all queries within that group.

To create an objective invariant to scale and attention scores distribution, we normalize the total cost by the cost incurred by an optimal ranking $\sigma^\star$. The final reward we optimize is normalized as $\mathcal{R}(\sigma_1, \dots, \sigma_n; \mathcal{X})/\mathcal{R}(\sigma_1^\star, \dots, \sigma_n^\star; \mathcal{X})$. In the experimental section, when we refer directly to $\mathcal{R}^b$, we always refer to its normalized version $\mathcal{R}^b(\sigma_1, \dots, \sigma_n; \mathcal{X})/\mathcal{R}(\sigma_1^\star, \dots, \sigma_n^\star; \mathcal{X})$.

### 3.2 PER-HEAD RL AGENT AND EFFICIENT TRAINING

A significant advantage of our method is its training efficiency. The agents are trained entirely offline, obviating the need for live LLM inference within the RL optimization loop. This is due to the fact that our usefulness score is function of true sequences, not generated sequences. First, we generate a dataset by running the base LLM on a training corpus. For each sample, we store the full sequence of queries, keys, and values. The attention matrices are not stored due to their prohibitive size.

During training, we sample a sequence and a cache size to be evicted from. We update the parameters $\theta$ of the agent using a policy gradient algorithm. Since our reward $\mathcal{R}(\sigma)$ is a terminal reward assigned only after the entire permutation $\sigma$ is generated, the objective is $J(\theta) = \mathbb{E}_{\sigma \sim \pi_\theta}[\mathcal{R}(\sigma)]$. We use the REINFORCE algorithm with a Leave-One-Out (RLOO) baseline (Williams, 1992; Ahmadian et al., 2024) to reduce variance. For an episode (permutation) $\sigma^k$ within a batch of $K$ episodes, the baseline $\bar{\mathcal{R}}$ is the average terminal reward of all other episodes in the batch. Considering the advantage, $\mathcal{R}(\sigma^k) - \bar{\mathcal{R}}$, the gradient is thus estimated as:

$$\nabla_\theta J(\theta) \approx \frac{1}{K} \sum_{k=1}^{K} \left[ (\mathcal{R}(\sigma^k) - \bar{\mathcal{R}}) \sum_{i=1}^{n} \nabla_\theta \log \pi_\theta(\sigma_i^k | \mathcal{X}) \right]. \tag{6}$$

We summarize the training in Alg. 1 and defer the further implementation details to Appendix A.2.

Training per-head agents from offline traces is highly scalable. At inference, the learned policy for each head ranks its KV entries, and the trailing entries are evicted based on the budget.

---

**Algorithm 1** RL Training Loop on Pre-Computed KV Traces

---

1: **Input:** Static dataset of pre-computed traces over $m$ sequences each with length $n_j, j \in [m]$ denoted as $\mathcal{X}^j = \{x_i^j\}_{i \in n_j}$ containing Q, K, V tensors.
2: **while** training not converged **do**
3:     Sample a data item $j \sim \texttt{Unif}(m)$ and a cache size $n \sim \texttt{Unif}(n_j)$.
4:     Sample $K$ permutations from the agent $\sigma_1^k, \ldots, \sigma_n^k \sim \pi_\theta(\sigma | x_1^j, \ldots, x_n^j), k \in [K]$
5:     Calculate reward $\mathcal{R}(\sigma_1^k, \ldots, \sigma_n^k; \mathcal{X}^j)$ using equation 4.
6:     Update $\theta$ using equation 6.
7: **end while**

---

## 4 EVALUATION

We hypothesize that a high-quality KV cache eviction policy can be trained entirely offline and deployed at inference time using only static token features (keys, values, and positions). To validate this, we conduct a comprehensive evaluation of KVP across multiple language modeling benchmarks and a wide range of cache budgets. Our results show that KVP consistently shows stronger downstream performance relative to existing heuristics, including methods that exploit privileged, query-specific information. We also provide ablation studies to analyze the design choices that enable this performance.

### 4.1 EXPERIMENTAL SETUP

**Base Model and Datasets.** Our experimental setup is centered on the `Qwen2.5-7B-Chat` model (Yang et al., 2024; Wang et al., 2024), a state-of-the-art transformer LLM. Please refer to Figure 6 in the Appendix for additional experiment on `Phi-4 14B` (Abdin et al., 2025). We train and evaluate our KVP agents using two distinct long-context benchmarks: *RULER-4k* (Hsieh et al., 2024), a synthetic dataset designed to probe long-context reasoning with sequences of approximately 4500 tokens; and *OASST2-4k*, a curated subset of the OpenAssistant Conversations Dataset (Köpf et al., 2023) featuring multi-turn dialogues of similar length. To assess generalization, we further perform zero-shot evaluation on four standard downstream benchmarks from the EleutherAI Language Model Evaluation Harness (Gao et al., 2024): *BoolQ* (Clark et al., 2019), a reading comprehension task framed as yes/no question answering; *ARC-Challenge* (Clark et al., 2018), a collection of science exam questions requiring multi-step reasoning; *MMLU* (Hendrycks et al., 2021), a benchmark testing expert-level knowledge across a wide range of domains; *HellaSwag* (Zellers et al., 2019), a commonsense sentence completion task; and *GovReport* (Bai et al., 2023), a summarization task from the LongBench benchmark for long context understanding. Importantly, KVP is not trained or fine-tuned on these tasks. Please refer to Appendix A.2 for further implementation details.

**Baselines.** We benchmark KVP against two main categories of KV cache management techniques: attention-based and attention-free. The first category includes state-of-the-art methods like *TOVA* (Oren et al., 2024) and *SnapKV* (Li et al., 2024). These methods leverage attention scores to identify important tokens, which introduces computational overhead by tying the eviction strategy to the attention calculation step. The second, more efficient category of attention-free baselines, to which our own KVP agents belong, operates independently of the attention mechanism. This group includes a *Random* eviction baseline, the recency-based *StreamingLLM* (Xiao et al., 2024), and several approaches that prune tokens based on their key embeddings. These include methods based on statistical patterns (*LagKV* (Liang et al., 2025)), vector similarity (*KeyDiff* (Park et al., 2025)), and L2 norm (*K-Norm* (Devoto et al., 2024)). To ensure a fair comparison, we adapt all baselines to our ranking-based framework by converting their binary keep/evict decisions into a full token permutation, enabling evaluation with consistent metrics.

**Budgeting and Compression Schedule.** For all online evaluations, we follow a consistent compression protocol. First, the entire context, including the final user message or question that prompts the generation, is processed by the LLM to populate the initial KV cache (the "prefill" stage). Second, the specified compression method is applied to this pre-populated cache to reduce its size to the target budget. Finally, the model generates the response autoregressively using the compressed cache. This

setup tests the ability to compress a large context before generation begins. To isolate the performance of the core ranking strategy, we apply a uniform compression budget across all heads and layers for all methods. This ensures a fair comparison focused purely on the quality of the eviction strategy itself, rather than on budget allocation heuristics. While head-specific or layer-specific budgeting is an orthogonal and promising direction for further performance improvements, our approach provides a clear and interpretable evaluation of the underlying policies.

**Inference Efficiency.** Our KVP agents make their ranking decisions using only the Key and Value vectors, and their position in the context. This makes KVP highly efficient, as it avoids re-computing attention scores. In contrast, attention-based baselines like TOVA and SnapKV are evaluated using the attention scores generated during the prefill stage, giving them access to information about how the final user message, for example, attends to the rest of the context. Our method is therefore benchmarked against baselines that have access to more direct, query-specific information at the time of compression. We highlight in all the figures the attention-based baselines with dashed lines.

**Absolute vs. Relative Cache Size.** A final methodological note concerns our use of cache size. Throughout our evaluation, we report performance as a function of absolute KV cache size (i.e., the number of tokens retained) rather than a relative compression ratio. This decision is motivated by practical application: practitioners operate under fixed memory constraints, making an absolute token budget a more direct and interpretable measure of resource cost. In contrast, a compression ratio's impact on memory is dependent on the initial context length, making it a less stable metric for cross-scenario comparison.

## 4.2 DOWNSTREAM PERFORMANCE

This section assesses the real-world efficacy of each compression method. We apply the strategies to a live LLM, reducing the KV cache to a target budget after the prefill before measuring performance on several downstream benchmarks.

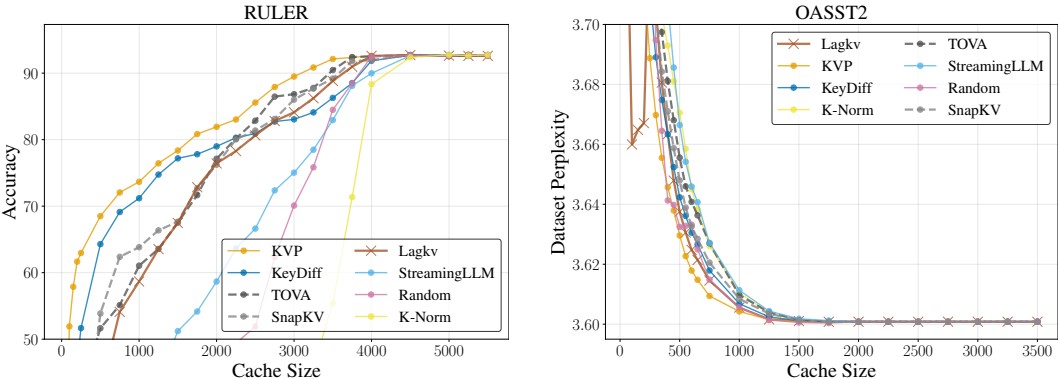

Figure 2: Overall accuracy on the RULER benchmark (Left) and perplexity on the OASST2-4k test set (Right), as a function of the absolute KV cache size. KVP achieves the highest accuracy and lowest perplexity across most of the possible cache sizes.

**RULER benchmark.** We evaluate performance on the RULER benchmark using its official text-based accuracy metric, which requires generating the correct answer for long-context reasoning tasks. This tests whether the methods preserve the specific, often non-local, information needed for complex problem-solving. Figure 2 (left) shows that KVP consistently outperforms all baselines, retaining higher accuracy as the cache budget shrinks. This result is a direct consequence of its learned policy: unlike heuristics that might discard old but crucial clues, KVP learns to identify and keep these high-value tokens, regardless of their position, enabling the model to succeed at the task. A detailed breakdown of performance across all RULER subtasks, along with their corresponding eviction error curves, is provided in Figure 12, further highlighting the robustness of our method. This result is particularly noteworthy given that RULER's structure heavily favors strategies that can use the final question to identify relevant context; an advantage held by attention-based baselines but not by KVP.

**OASST2 benchmark.** We evaluate the efficacy of KV cache compression by its impact on perplexity (PPL), a measure of the model's next-token prediction capability. An effective compression method should minimize PPL degradation. As shown in Figure 2 (right), KVP consistently achieves lower perplexity than baselines that rely solely on KV embeddings. Furthermore, it often outperforms methods that use additional information across nearly all cache sizes. In contrast, heuristic approaches like *StreamingLLM* exhibit a sharp increase in PPL as the cache budget decreases, confirming the brittleness of their fixed-rule strategies. The superior performance of KVP demonstrates a direct link between our training objective and the preservation of the model's downstream capabilities. These findings are further corroborated by results on the RULER benchmark (Appendix Figure 9), where KVP maintains a similar advantage.

**Generalization to other benchmarks.** LLMs are widely regarded as general-purpose computational systems and their utility depends on robust performance in out-of-domain scenarios. Even though KVP enables efficient training of domain specific policy using only unlabelled dataset of sequences, we evaluate its zero-shot generalization performance on a set of standard downstream tasks. Specifically, we test agents trained on RULER ($KVP^R$) and *OASST2* ($KVP^S$) on *ARC-Challenge*, *BoolQ*, and the *GovReport* task from the LongBench benchmark, reporting performance at varying KV cache sizes. Importantly, the "prefill" stage does not include the question in *BoolQ* and *GovReport*, so as to better reflect real-world scenarios where a given text is compressed for different questions not known in advance. Results on *MMLU* and *HellaSwag* are instead presented in Appendix A.4. This evaluation is designed to assess whether policies optimized for long-context efficiency maintain performance on short and long-context benchmarks that probe factual knowledge, multi-step reasoning, and commonsense inference.

The results, presented in Figures 3 and 4, demonstrate that KVP consistently achieves competitive performance. Across the diverse benchmarks, both $KVP^R$ and $KVP^S$ rank at or near the top, outperforming most heuristic baselines and closely tracking the accuracy of the uncompressed model, demonstrating minimal degradation at different cache size reduction. While both agents generalize effectively, some specialization emerges: $KVP^S$ (trained on conversational data) shows an advantage on the natural question answering task *BoolQ* and the summarization task *GovReport*, whereas $KVP^R$ (trained on synthetic reasoning sequences) performs best on the expert-knowledge benchmark *MMLU* as shown in Figure 10 in Appendix A.4. In contrast to the baselines, whose relative ranking is inconsistent across benchmarks, these results highlight that the adaptive policy learned by KVP generalizes far beyond their training domain and context lengths. This success preserves the model's core capabilities, confirming that KVP is not a specialized tool with significant trade-offs, but a robust technique that can be enabled by default to provide memory savings while minimizing degradation to the model's general utility.

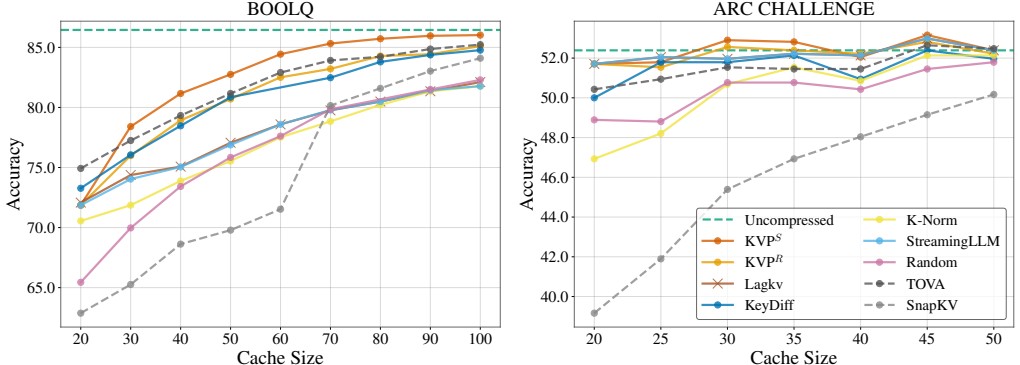

Figure 3: Average test accuracy on (Left) BoolQ and (Right) ARC Challenge as a function of KV cache size. Higher is better.

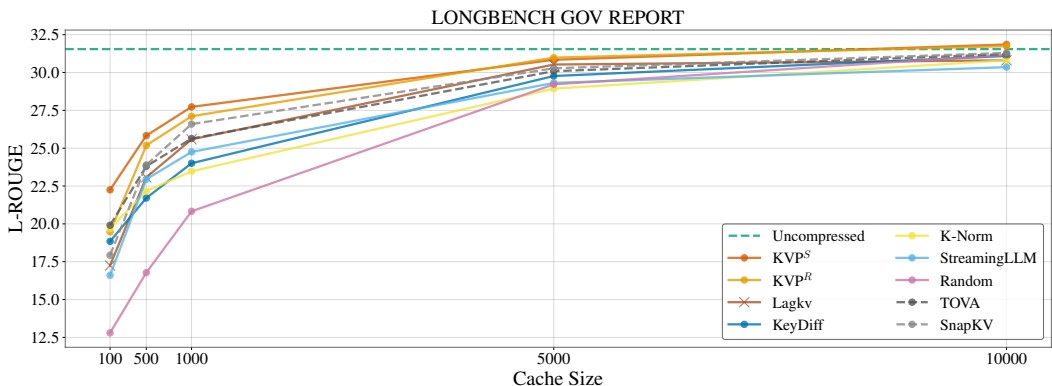

Figure 4: Average test L-Rouge score on the GovReport as a function of KV cache size. Higher is better.

## 4.3 ABLATIONS

To validate the core design choices of KVP, we conduct two key ablation studies. First, we analyze our proposed reward function, as defined in Equation (4). Second, we study our use of RL instead of supervised surragates of sorting.

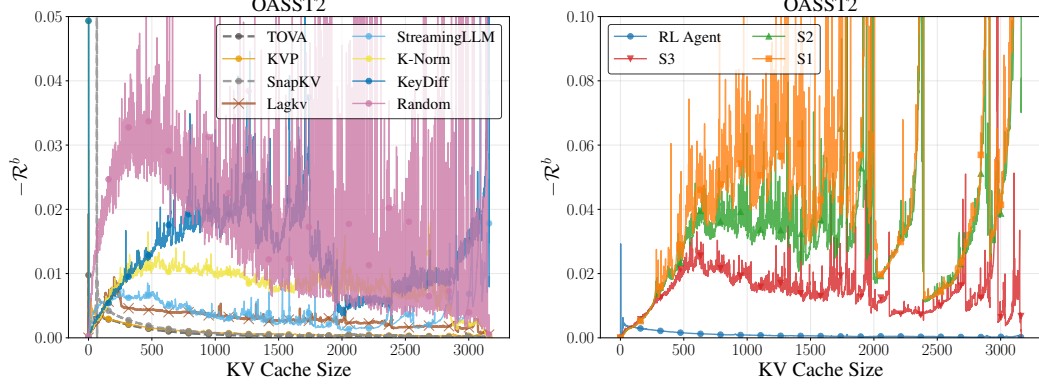

(a) **Effectiveness of the reward:** KVP performs on par with attention-aware methods (dashed lines) despite not using their privileged information by learning to predict future attentions. A full visualization for all the layers is in Figures 13 and 14.

(b) **RL vs Supervised Surrogates of Sorting:** The RL agent successfully reaches low cost $(-\mathcal{R}^b)$ while the supervised surrogate (Blondel et al., 2020) fails. We tested the supervised approach with three learning rates: $5 \times 10^{-6}, 5 \times 10^{-5}$ and $5 \times 10^{-4}$.

Figure 5: Cost per-budget $(-\mathcal{R}^b)$ on the OASST2 test set for a representative head (layer 10, head 0).

**Is the reward function $\mathcal{R}$ effective?** To confirm that our reward $\mathcal{R}$ enables effective learning of eviction policies, we measure the negative per-budget reward, $-\mathcal{R}^b$, on unseen test data. This value corresponds to the total future importance of the tokens evicted at a given budget $b$ which our agent is trained to minimize across all budgets simultaneously.

The results in Figure 5a show a clear separation between methods: while attention-aware methods that use query information at inference time (dashed lines) form a distinct, low-loss cluster, heuristics without attention scores form a high-loss cluster. This is rather expected as the target, future attention scores, is highly related to attention. Crucially, KVP performs within this top-tier group, achieving a loss comparable to methods like *TOVA* and *SnapKV* without using their privileged information. This demonstrates that our offline RL training successfully distills the principles of attention-based ranking into an efficient, static policy. Furthermore, the analysis reveals the importance of policy specialization across heads. A fixed heuristic like *StreamingLLM* can be effective for certain heads (see layer 22, head 0 in Figure 14) but detrimental for others (see layer 19, head 0 in Figure 13). Even

*KeyDiff*, a strong attention-free heuristic, exhibits hard failure modes (see layer 2, head 0 in Figure 14). This result, together with a comprehensive sweeps across all layers and heads in Figures 13 and 14, confirms that KVP learns diverse patterns that cannot be captured with a single heuristic.

**Is RL necessary?** We ablate the necessity of using RL by comparing it to a supervised learning baseline using differentiable surragate of sorting (Blondel et al., 2020). This baseline uses the same data and network architecture but replaces the policy gradient objective with a length-normalized Mean Squared Error loss between predicted soft ranks (from a differentiable sorter (Blondel et al., 2020)) and the ground-truth ranks. As shown in Figure 5b, this leads to poor performance: the supervised agent fails to learn an effective policy, evidenced by its high and unstable cost curve. In contrast, the RL agent's cost is low and stable confirming that RL is better suited for learning KV cache policies.

We conjecture that the KV cache eviction task is inherently sparse: only a small fraction of tokens are critical for future generation. This sparsity poses a fundamental challenge for supervised methods using differentiable sorting surrogates. Such methods face a trade-off: a low temperature provides vanishing gradients, while a high temperature yields dense but biased gradients that incorrectly disperse importance across irrelevant tokens. Policy gradient methods bypass this issue by providing an unbiased learning signal through the non-differentiable sort operation. This allows for direct credit assignment, concentrating the learning signal on the few ranking decisions that matter.

## 5 CONCLUSIONS AND FUTURE WORK

In this work, we introduced a new paradigm for KV cache eviction by reframing it as a learnable sorting problem. We proposed KVP, a system of lightweight, per-head policies trained to rank cache entries by their predicted future utility. The core of our method is a reward signal based on the ranking quality across all possible memory budgets and its efficient inference-free optimization via reinforcement learning. This approach allows for fine-grained, adaptive memory control without requiring architectural changes or costly live LLM inference during training.

Our evaluations on long-context benchmarks demonstrate that KVP consistently outperforms strong heuristic baselines, achieving lower perplexity degradation and higher task accuracy under tight memory constraints. This validates our central hypothesis: directly learning a specialized sorting policy is a more powerful and robust strategy than relying on handcrafted, backward-looking proxies for token importance. By preserving model fidelity more effectively, KVP establishes a promising direction for building more efficient and capable LLM inference systems.

Future work could extend this framework in several exciting directions. A natural next step is to develop an adaptive budget allocation policy that dynamically assigns memory where it is most needed across heads and layers. Furthermore, while KVP is highly efficient, it does not model interdependencies between heads. A powerful extension would be to explore joint, end-to-end optimization of all policies, which could capture these complex dynamics at the cost of requiring online training.

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

# A  APPENDIX

## A.1  MISSING PROOF OF PROPOSITION 1

*Proof.* Since $|S^\star_{b+1}| = b+1$ and $|S^\star_b| = b$ with $S^\star_b \subset S^\star_{b+1}$, the set difference $S^\star_{b+1} \setminus S^\star_b$ is nonempty. By uniqueness of $S^\star_{b+1}$, this difference must contain exactly one element; otherwise, there would exist two distinct $(b+1)$-subsets strictly between $S^\star_b$ and $S^\star_{b+1}$, contradicting uniqueness. Hence we can define a sequence of distinct elements

$$x_{\sigma_1} \in S^\star_1, \qquad x_{\sigma_{b+1}} \in S^\star_{b+1} \setminus S^\star_b \quad (b = 1, \ldots, n-1).$$

By construction, for each $b$ we have

$$S^\star_b = \{x_{\sigma_1}, \ldots, x_{\sigma_b}\}.$$

Now define a total order (ranking) $\pi$ on $[n]$ by setting

$$\pi(x_{\sigma_b}) = b \quad (b = 1, \ldots, n).$$

This is a bijection $\pi : [n] \to [n]$, and its top-$b$ prefix is precisely $\{x_{\sigma_1}, \ldots, x_{\sigma_b}\} = S^\star_b$. Therefore,

$$S^\star_b = \{\, i \in [n] : \pi(i) \leq b \,\} \quad \text{for all } b,$$

as claimed.

Equivalently, given $\pi$ we may define a scoring function consistent with the order, for instance

$$s(x_{\sigma_k}) = n - k \quad (k = 1, \ldots, n),$$

which is strictly decreasing in $k$. Then the top-$b$ elements according to $s$ are exactly $\{x_{\sigma_1}, \ldots, x_{\sigma_b}\} = S^\star_b$. □

## A.2 IMPLEMENTATION DETAILS

Our KVP agents are lightweight 2-layer MLPs with 256 hidden units, trained using the RLOO algorithm (Ahmadian et al., 2024) as described in Section 3. Following the Grouped-Query Attention (GQA) architecture of `Qwen2.5-7B-Chat`, we train a separate agent for each of the 4 KV heads across all 28 layers, yielding 112 specialized agents. Each agent contains approximately 600K parameters.

The agents are optimized to maximize the reward signal in Equation (4), which encourages retention of tokens with high future utility across all cache budgets. During inference, the learned policy ranks all tokens except the first 4 and last 16, which are always retained. Token eviction is emulated via custom attention masks in FlexAttention, preventing the model from attending to pruned tokens during generation.

For training stability, we apply gradient clipping (maximum norm of 5) and normalize advantages by their mean and standard deviation, without entropy regularization. We use AdamW with a learning rate of $5 \times 10^{-5}$, following a cosine schedule with 100-step linear warmup (start factor 0.01) that decays to $1 \times 10^{-6}$. Each agent trains for 4,000 steps on pre-computed activations.

## A.3 TRAINING AND MEMORY OVERHEAD

The training process for our KVP agents is designed to be highly efficient, imposing minimal computational and storage overhead. Agents are trained on a small dataset of pre-computed activation traces; for our experiments, we used approximately 6,000 samples from the RULER dataset and 4,500 from the OASST2 dataset. Data generation represents the primary one-time cost of this pipeline, requiring a single forward pass over these training samples to collect and store on disk the Query, Key, and Value embeddings for the entire generation sequence for each KV cache. This step incurs a computational cost approximately equivalent to standard inference on the dataset. Each per-head agent is a small MLP with approximately 650K parameters, resulting in a checkpoint size of only 2.6MB.

The agent training completes in less than 30 minutes on a single node of 8 NVIDIA H100 GPUs. We note that this training time is achieved without extensive hyperparameter tuning or code optimizations for speed, suggesting that further significant speed-ups are possible. These factors highlight the low computational footprint of our offline training framework.

**FLOPs estimation**. To assess computational cost independent of hardware optimization (e.g., kernel fusion), we quantify the FLOPs overhead per token:

- Autoregressive Generation (Throughput). Zero Overhead. In our setting, the KV cache is compressed only once after prefill. This means KVP introduces no additional FLOPs during the decoding phase. Consequently, its throughput is identical to any other eviction method at the same cache budget. KVP's contribution is achieving superior accuracy for that given level of throughput.

- Prefill Latency. Cost: 14.00 GFLOPs $\rightarrow$ 14.15 GFLOPs. The overhead is strictly confined to the prefill stage. Based on the standard approximation of $2 \times N_{params}$ FLOPs per token:
  - Base model (Qwen-7B): 14.00 GFLOPs/token.
  - KVP Overhead: Our 112 agents (0.65M params each) add 72.8M parameters, contributing an additional 0.15 GFLOPs/token.
  - Total: The prefill cost becomes 14.15 GFLOPs/token, a marginal increase of 1%.

This quantitative breakdown demonstrates that the computational footprint of KVP is negligible compared to the backbone model.

## A.4 ADDITIONAL RESULTS

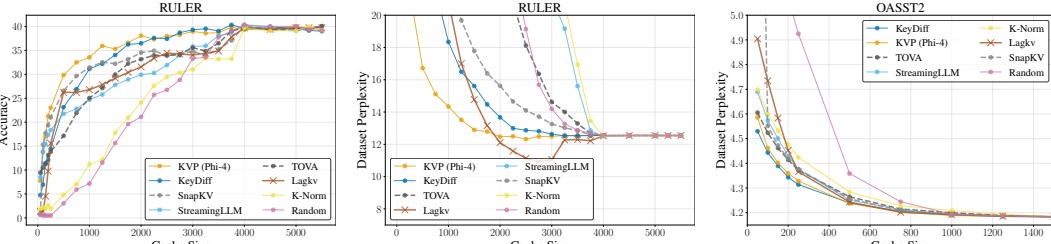

Figure 6: Evaluation of our KVP eviction strategy on the `Phi-4 14B` model. The results demonstrate that our learning framework generalizes effectively to a different model architecture. While KVP's performance remains consistently strong, the relative performance of heuristic baselines changes significantly between Qwen 2.5 and Phi-4. This provides strong evidence that their effectiveness is model-dependent, unlike our learned approach. **(left)** KVP achieves the high accuracy on the RULER benchmark, successfully adapting its learned policy where attention-based heuristics (SnapKV, TOVA) fail due to the task's nature. **(center)** Perplexity on the RULER test set, where KVP consistently outperforms baselines. **(right)** Perplexity on the OASST2-4k test set, confirming KVP's good performance.

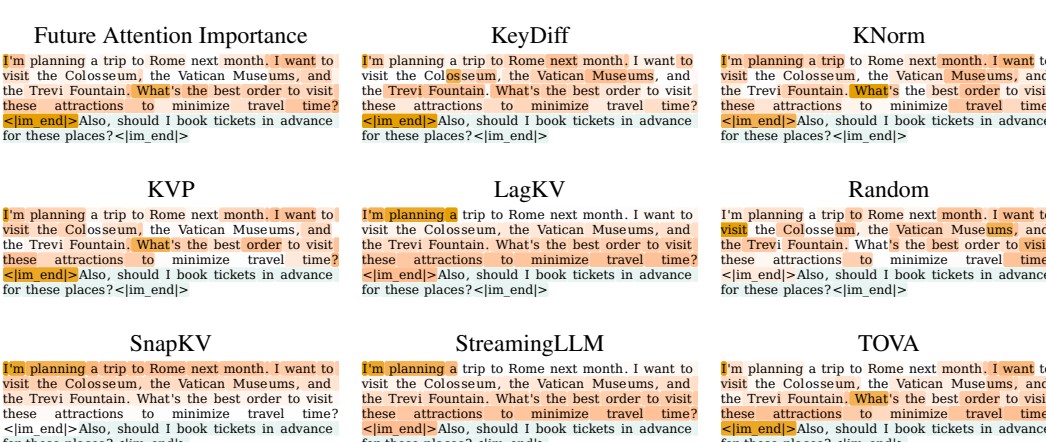

Figure 7: All strategies compared on the qualitative example shown in Figure 1. The attention scores considered are from layer 12 head 0. Another qualitative example in Figure 11 and a failure case in Figure 8

### Future Attention Importance

If a store offers a 20% discount on a jacket and the sale price is $80, what was the original price? Please calculate the answer and let me know what you get.<|im_end|>The original price was $100. Since the store took 20% off, the sale price represents the remaining 80% of the original price (100% - 20% = 80%). To find the full price (100%), you divide the sale price by 0.80.<|im_end|>

### KeyDiff

If a store offers a 20% discount on a jacket and the sale price is $80, what was the original price? Please calculate the answer and let me know what you get.<|im_end|>The original price was $100. Since the store took 20% off, the sale price represents the remaining 80% of the original price (100% - 20% = 80%). To find the full price (100%), you divide the sale price by 0.80.<|im_end|>

### KNorm

If a store offers a 20% discount on a jacket and the sale price is $80, what was the original price? Please calculate the answer and let me know what you get.<|im_end|>The original price was $100. Since the store took 20% off, the sale price represents the remaining 80% of the original price (100% - 20% = 80%). To find the full price (100%), you divide the sale price by 0.80.<|im_end|>

### KVP

If a store offers a 20% discount on a jacket and the sale price is $80, what was the original price? Please calculate the answer and let me know what you get.<|im_end|>The original price was $100. Since the store took 20% off, the sale price represents the remaining 80% of the original price (100% - 20% = 80%). To find the full price (100%), you divide the sale price by 0.80.<|im_end|>

### LagKV

If a store offers a 20% discount on a jacket and the sale price is $80, what was the original price? Please calculate the answer and let me know what you get.<|im_end|>The original price was $100. Since the store took 20% off, the sale price represents the remaining 80% of the original price (100% - 20% = 80%). To find the full price (100%), you divide the sale price by 0.80.<|im_end|>

### Random

If a store offers a 20% discount on a jacket and the sale price is $80, what was the original price? Please calculate the answer and let me know what you get.<|im_end|>The original price was $100. Since the store took 20% off, the sale price represents the remaining 80% of the original price (100% - 20% = 80%). To find the full price (100%), you divide the sale price by 0.80.<|im_end|>

### SnapKV

If a store offers a 20% discount on a jacket and the sale price is $80, what was the original price? Please calculate the answer and let me know what you get.<|im_end|>The original price was $100. Since the store took 20% off, the sale price represents the remaining 80% of the original price (100% - 20% = 80%). To find the full price (100%), you divide the sale price by 0.80.<|im_end|>

### StreamingLLM

If a store offers a 20% discount on a jacket and the sale price is $80, what was the original price? Please calculate the answer and let me know what you get.<|im_end|>The original price was $100. Since the store took 20% off, the sale price represents the remaining 80% of the original price (100% - 20% = 80%). To find the full price (100%), you divide the sale price by 0.80.<|im_end|>

### TOVA

If a store offers a 20% discount on a jacket and the sale price is $80, what was the original price? Please calculate the answer and let me know what you get.<|im_end|>The original price was $100. Since the store took 20% off, the sale price represents the remaining 80% of the original price (100% - 20% = 80%). To find the full price (100%), you divide the sale price by 0.80.<|im_end|>

Figure 8: A qualitative comparison of all strategies, similar to Figure 1. In this particular failure case, the attention scores from layer 10, head 0, show that our proposed KVP strategy struggles to fully capture the ground-truth future attention pattern.

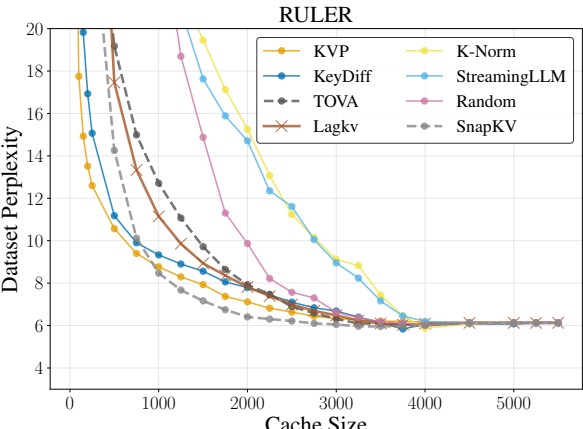

Figure 9: Perplexity (PPL) as a function of KV cache size. KVP achieves highly competitive perplexity, performing on par with or better than the leading baselines at most cache sizes and significantly outperforming other methods. This result is particularly notable given that RULER's structure, which includes random sentences preceding a final question, heavily advantages methods that can isolate tokens relevant to that question.

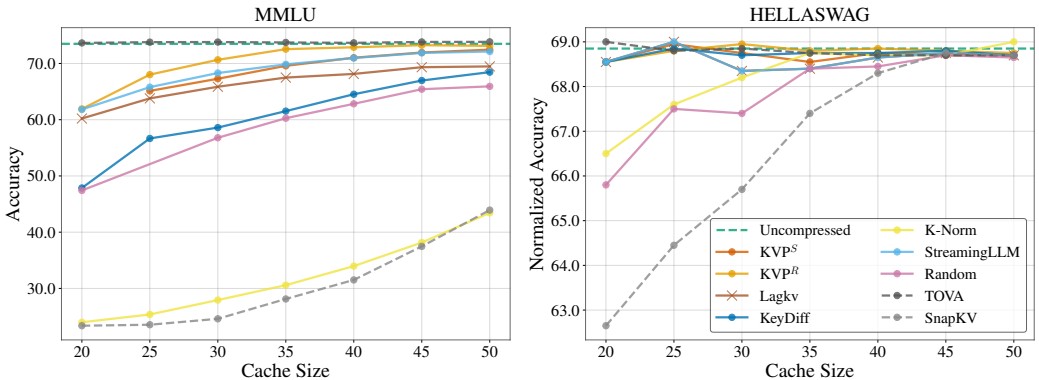

Figure 10: (Left) Average test accuracy on MMLU and (Right) average normalized accuracy on Hellaswag as a function of KV cache size. Higher is better.

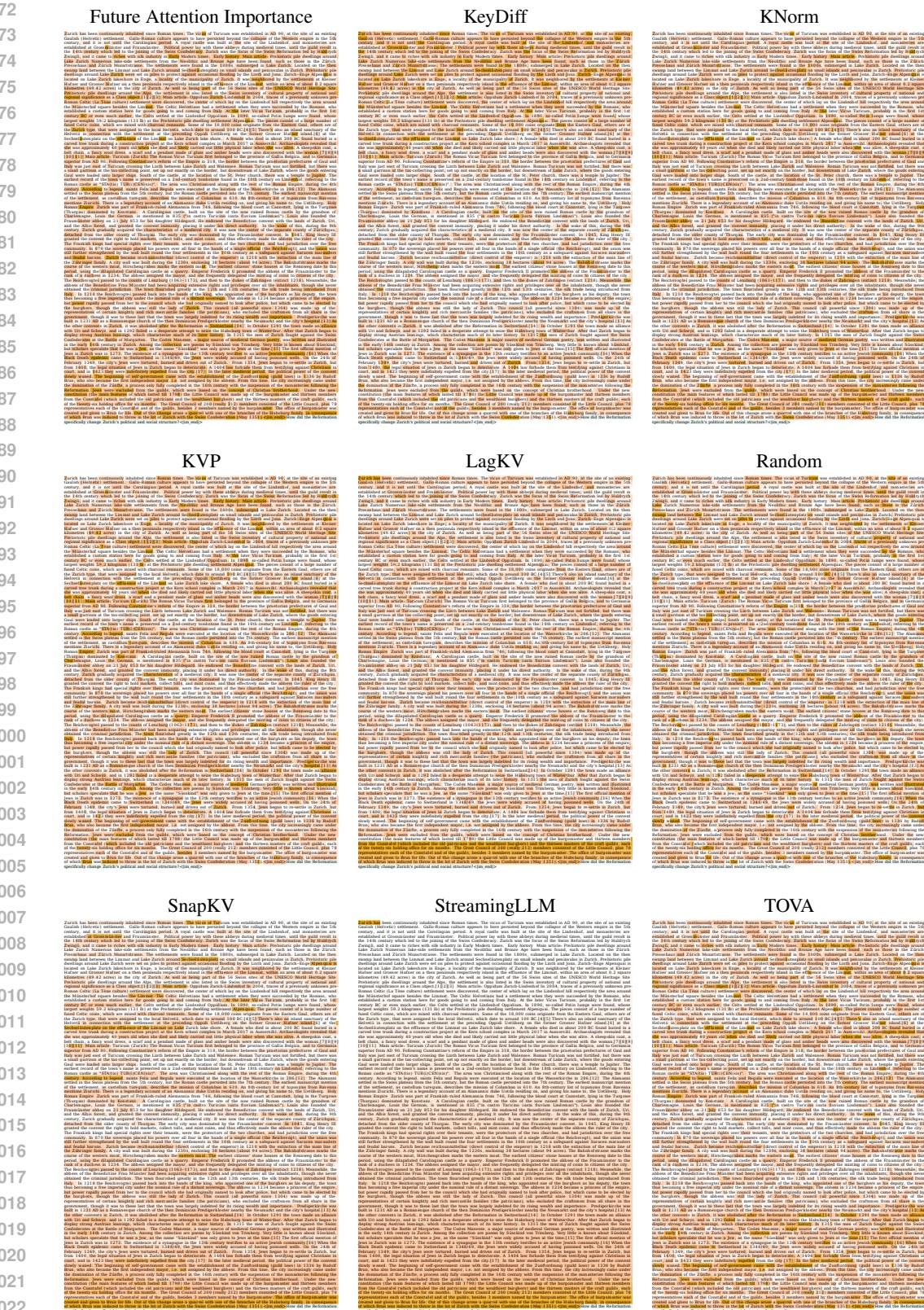

Figure 11: All strategies compared on a long qualitative example. The attention scores considered are from layer 19 head 0.

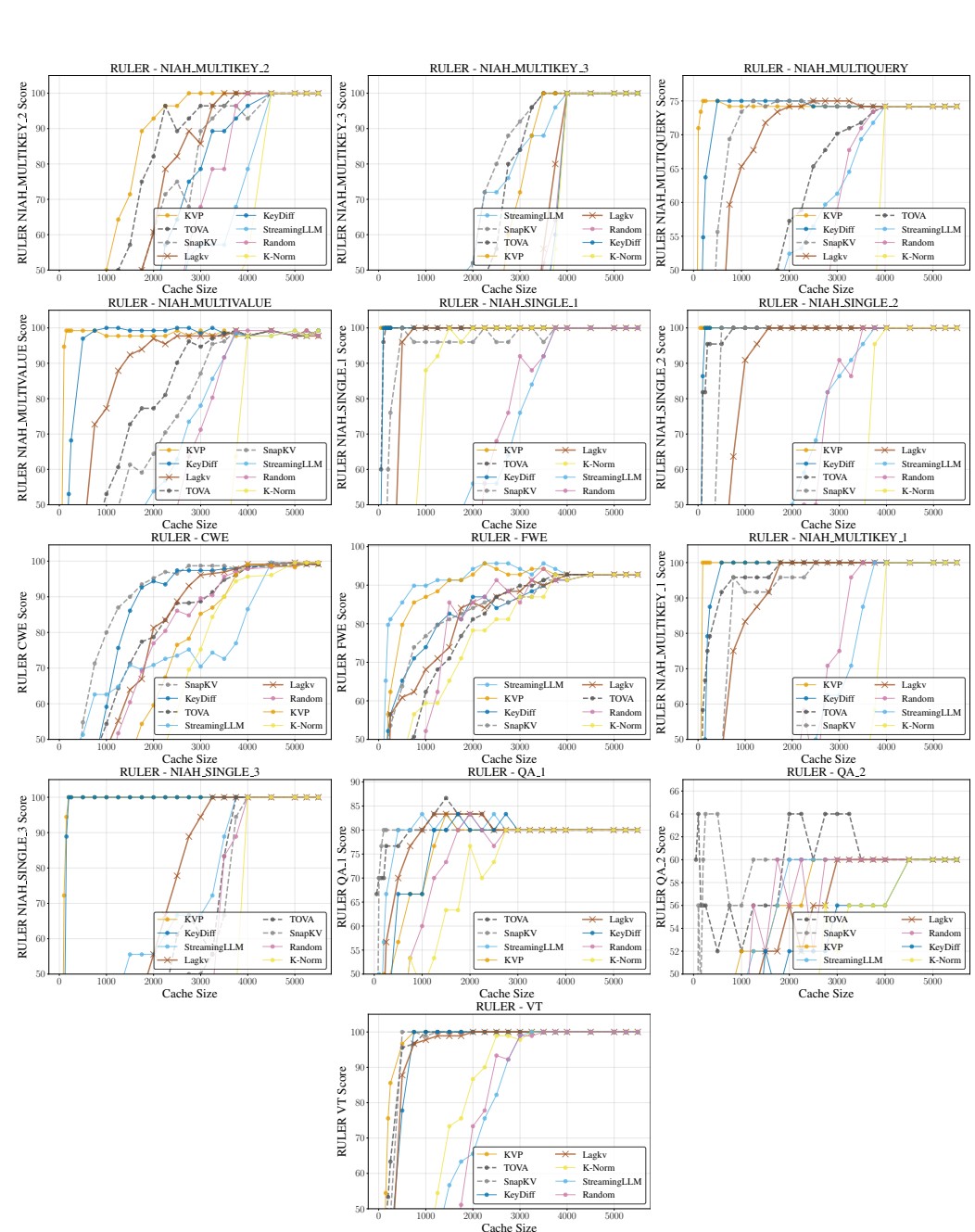

Figure 12: Per-task accuracy on all RULER subtasks as a function of absolute KV cache size.

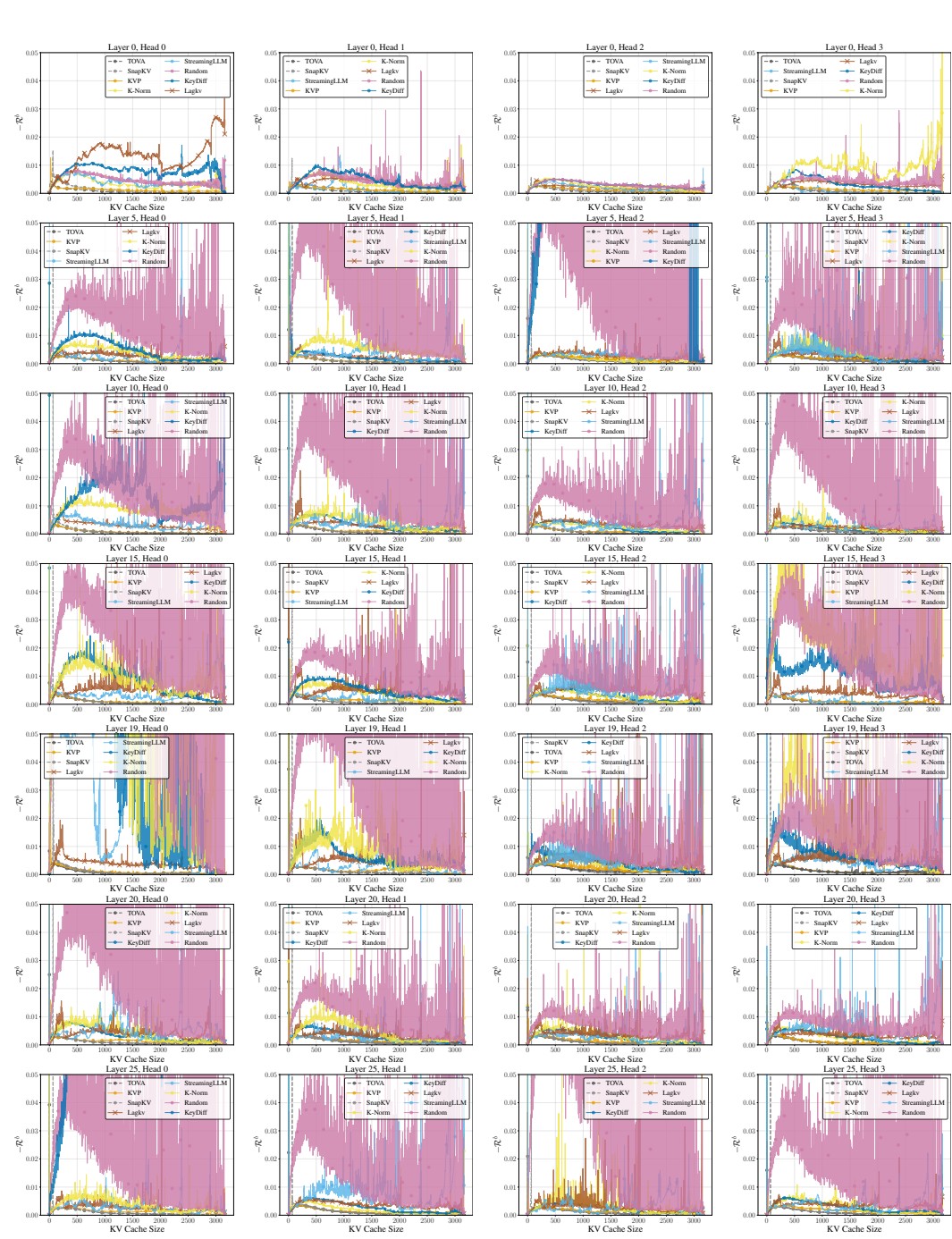

Figure 13: Cost $(-\mathcal{R}^b)$ for all strategies across a selection of layers (rows) and all available heads (columns) on the OASST2 test set. The plots show that the relative performance of different strategies varies significantly across heads, highlighting the benefit of learning specialized per-head policies. Lower is better.

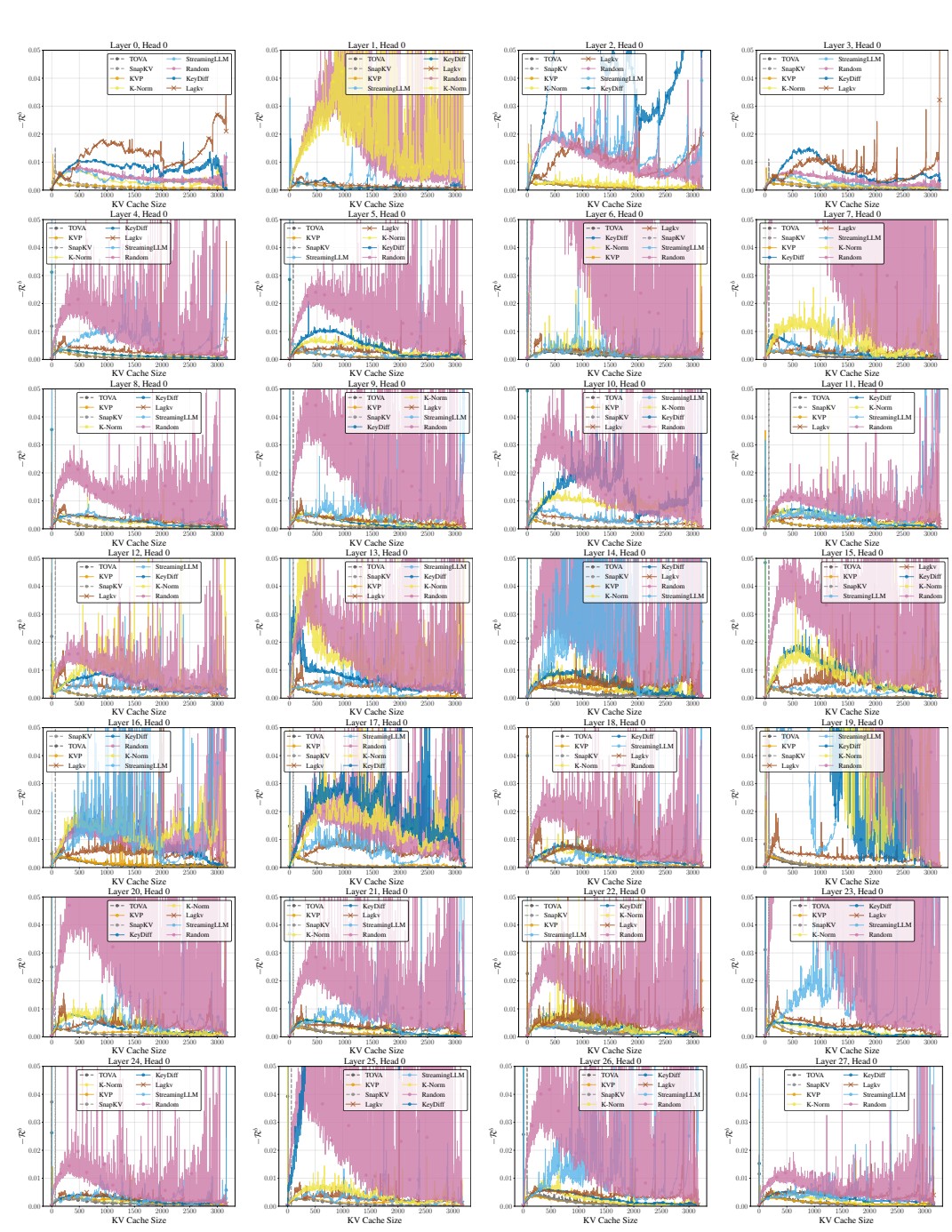

Figure 14: Cost $(-\mathcal{R}^b)$ for all strategies on head 0 across all 28 layers of the model, evaluated on the OASST2 test set. This visualization shows how the effectiveness of different non attention-aware eviction heuristics changes with model depth, whereas the learned KVP policy remains consistently effective. Lower is better.

