# OpenReview forum: "Learning to Evict from Key-Value Cache"
_ICLR.cc/2026/Conference — Submitted to ICLR 2026_

### Official Review · Reviewer_DJnn · 2025-10-24

**Soundness:** 2
**Presentation:** 2
**Contribution:** 2
**Rating:** 2
**Confidence:** 5

**Summary:**

This paper introduces KV Policy, a reinforcement learning framework for KV cache eviction in LLMs. Instead of using heuristic policies (recency, attention score, etc.), KVP learns to rank tokens by predicted future utility. Each attention head in the LLM is paired with a lightweight RL agent trained offline on pre-computed KV traces using only keys, values, and positional embeddings. The reward function evaluates the ranking quality across all cache budgets without additional inference. Experiments on long-context benchmarks (RULER, OASST2) and zero-shot downstream tasks (BoolQ, ARC-Challenge, MMLU, HellaSwag) show that KVP achieves consistently better accuracy and lower perplexity than heuristic and attention-based baselines such as SnapKV, TOVA, and StreamingLLM.

**Strengths:**

1. The use of per-head lightweight RL agents is well-motivated and efficiently parallelizable.

2. The authors include careful ablations isolating the contribution of the RL objective and the reward design.

**Weaknesses:**

1. While the paper claims that KVP introduces minimal overhead, there are no quantitative measurements of inference latency or throughput. Reporting actual wall-clock runtimes or speedups relative to heuristic and attention-based baselines would strengthen the empirical evaluation.

2. The paper claims offline efficiency, but training 112 separate agents still requires substantial GPU resources. A clearer cost-benefit analysis would help.

3. Because KVP requires a dedicated offline training phase, comparing it only with training-free heuristic baselines is not entirely fair. It would strengthen the evaluation to include comparisons with other training-based or learned compression methods, such as Gisting Token (https://arxiv.org/abs/2509.15763) or Activation Beacon (https://arxiv.org/abs/2401.03462).

4. Experiments are limited to one base model (Qwen2.5-7B-Chat). It would strengthen the paper to demonstrate that the learned policies generalize to other architectures (e.g., Llama, Mistral).

5. Since the core motivation is to reduce memory and latency, experiments on truly long-context settings (e.g., 10k–100k tokens or more) are necessary. The current benchmarks (RULER and OASST2-4k) do not fully test KVP’s scalability under extreme context lengths, limiting the conclusions about its real-world applicability.

6. The generalizability of KVP remains uncertain. It would be important to assess how sensitive the learned policies are to the choice of offline training data. For example, would a policy trained on conversational datasets transfer effectively to domains such as code generation or complex reasoning tasks?

7. KVP operates offline and applies a fixed learned policy per head. This limits adaptivity during runtime when generation dynamics may differ from training data distributions.

**Questions:**

See the limitations.

---

> ### Author Response · Authors · 2025-11-26
>
> We sincerely thank the reviewer for their thorough summary and constructive feedback. We are encouraged that the reviewer recognizes the well-motivated design of our per-head RL agents and the careful ablation studies. In response, we have added two significant new experiments: (1) a full evaluation on the Phi-4 14B model to demonstrate scalability across architectures (Figure 6), and (2) a new zero-shot generalization experiment on the GovReport subtask of LongBench to test performance on very long contexts (Figure 4).
>
> Below, we address each of the weaknesses raised by the reviewer.
>
> **W1: Quantitative measurements of inference latency or throughput.**
> We thank the reviewer for this crucial point. To assess computational cost independent of hardware optimization (e.g., kernel fusion), we quantify the exact FLOPs overhead per token.
> 1. Autoregressive Generation (Throughput). Zero Overhead. In our setting, the KV cache is compressed only once after prefill. This means KVP introduces no additional FLOPs during the decoding phase. Consequently, its throughput is identical to any other eviction method at the same cache budget. KVP's contribution is achieving superior accuracy for that given level of throughput.
> 2. Prefill Latency. Cost: 14.00 GFLOPs ->  14.15 GFLOPs. The overhead is strictly confined to the prefill stage. Based on the standard approximation of $2 * N_{params}$ FLOPs per token:
>     - Base model (Qwen-7B): 14.00 GFLOPs/token.
>     - KVP Overhead: Our 112 agents (0.65M params each) add 72.8M parameters, contributing an additional 0.15 GFLOPs/token.
>     - Total: The prefill cost becomes 14.15 GFLOPs/token, a marginal increase of ~1%.
>
> This quantitative breakdown demonstrates that the computational footprint of KVP is negligible compared to the backbone model.
>
> **W2: Offline training cost and cost-benefit analysis.**
> This is an important practical consideration. We have added a detailed breakdown of the training costs in the appendix in A.3. The training for KVP is designed to be a small, one-time investment for a significant and generalizable performance gain.
> - Agent Size: Each per-head agent is a very small MLP with only ~650K parameters. For the 112 agents of the Qwen-7B model, the total storage is negligible compared to the base model (each agent checkpoint is 2.6MB).
> - Training Data: Training is performed on pre-computed offline traces (~6,000 samples for RULER, ~4,500 for OASST2). Generating these traces requires a one-time inference pass over the dataset, which is a standard data preparation cost.
> - Training Time: The training process  of each agent completes in less than 30 minutes on 8 H100 GPUs.
>
> This low computational footprint demonstrates that our offline training framework is a practical and scalable solution, since each agent can be trained in parallel and independently from the others.
>
> **W3: Comparison with other training-based methods.**
> We thank the reviewer for suggesting these highly relevant works. We agree that positioning KVP in the context of other learned methods is important. Our rationale for focusing on heuristic and simple attention-based baselines was to establish a clear, apples-to-apples comparison for the core task of KV cache eviction.
> Methods like UniGist and Activation Beacon are conceptually different and, we argue, complementary. They focus on creating a compressed summary token or condensed activations to represent past context, rather than learning a fine-grained ranking over all tokens for eviction at any arbitrary budget. KVP's goal is to produce a universal ranking that is effective across the full spectrum of compression ratios.
> We see these approaches as orthogonal. For instance, KVP could be used to decide which tokens are least important and can be summarized by a Gisting Token, representing a powerful potential synergy. We have updated our related work section to clarify this distinction and to better contextualize KVP within the broader landscape of learned compression techniques, highlighting these exciting future research directions. Furthermore, UniGist is concurrent work, released only a few days before our submission, which further explains why it was not included in our initial experimental comparisons.
>
> [1/2]

---

> > ### Author Response · Authors · 2025-11-26
> >
> > **W4: Experiments are limited to one base model.**
> > We completely agree that demonstrating generalization across different model architectures is critical. To address this, we have conducted a full new set of experiments on a different and larger model Microsoft's Phi-4 14B. Importantly, we used the same hyperparameters as in the Qwen2.5 experiments, further highlighting the strong generalizability and robustness of our approach across model families.
> > Our new results in Figure 6 show two key findings:
> > * KVP's performance remains consistently strong, outperforming all baselines on Phi-4 and confirming that our learning framework generalizes effectively to a different architecture.
> > * The relative performance of the heuristic baselines changes significantly between Qwen-7B and Phi-4. For example, a strategy that works well on one model may perform poorly on the other.
> >
> > This provides powerful evidence for the main benefit of KVP: it offers a robust framework that learns to adapt to the unique attention patterns of any given model, freeing developers from relying on hand-crafted heuristics that may not generalize.
> >
> > **W5&6: Sensitive to different offline training data and performance on "truly long-context settings”.**
> > Our generalization section (Sec. 4.2; Figs. 3, 10) *already* evaluates zero-shot transfer across diverse and out-of-domain benchmarks, including ARC-Challenge, BoolQ, MMLU, HellaSwag. As shown in Figures 3, policies trained on RULER (synthetic reasoning) and OASST2 (conversational data) consistently rank at or near the top across tasks spanning factual QA, multi-step reasoning, and expert knowledge. This demonstrates that KVP is not tied to any specific training distribution: KVP^R and KVP^S transfer effectively and retain high performance even when the prefill contains no task information (as in BoolQ), faithfully reflecting real-world deployment.
> > To further address the reviewer's concerns about both generalizability to "truly long-context settings," we conducted a new, challenging zero-shot experiment on the entire GovReport subtask from the LongBench benchmark. The results (Figure 4) show that KVP agents, trained on either RULER or OASST2, substantially outperform all baselines. This is a crucial finding: it simultaneously confirms that KVP generalizes to unseen domains and scales effectively to much longer contexts than seen during training. This combination of results provides powerful evidence that KVP generalizes far beyond its training domain and that its learned token-ranking policy is robust rather than specialized.
> >
> >
> > **W7: Offline policy limits runtime adaptivity.**
> > The reviewer correctly identifies the trade-off between a static offline policy and a dynamic online one. This was a deliberate design choice to prioritize inference efficiency. The core goal of KVP is to create a policy that can be applied with virtually no latency overhead during autoregressive generation. Online methods that adapt to runtime dynamics (e.g., by using the current query to re-evaluate token importance) inherently introduce significant latency. Our strong empirical results, also in the LongBench GovReport task, where KVP outperforms or is highly competitive with attention-aware baselines like SnapKV and TOVA, provide compelling evidence that our offline-trained, forward-looking policy is a highly effective and efficient proxy for true runtime utility. In fact, our analysis in the appendix (Figures 9 and 10) already shows that our offline policy's predictions of future attention scores are quantitatively comparable to those of attention-aware baselines. This demonstrates the power of our learning framework to capture a robust utility signal without incurring the high-cost of using attention scores.
> >
> > We believe these new experiments, quantitative analyses, and clarifications substantially strengthen our paper. We thank the reviewer again for their invaluable guidance and hope our revisions have addressed their concerns.
> >
> > [2/2]

---

### Official Review · Reviewer_xyfd · 2025-10-26

**Soundness:** 2
**Presentation:** 4
**Contribution:** 3
**Rating:** 4
**Confidence:** 4

**Summary:**

This submission investigates the problem of KV cache management in LLMs and proposes a method called KVP which adopts Reinforcement Learning to learn a strategy minimizing the future value of tokens to be evicted from the cache. While this is an interesting, novel approach, the submission does not make a compelling case that KVP improves the performance of SOTA methods in a significant way.

**Strengths:**

The formulation of KV cache eviction as a learning problem is original. The authors prove that, under two reasonable assumptions, the subset selection problem can be reduced to a ranking problem, which can then be formulated as an RL problem.

It is interesting that the policy requires only the keys, values, and their positions as input and no attention information. It is a strength of KVP that it can be pretained and does not incur any overhead at inference time.

I appreciate the ablation study demonstrating that supervised learning does not work and RL is necessary.

The paper is written clearly.

**Weaknesses:**

The experiments show that KVP achieves the best accuracy or perplexity on RULER and OASST2 for most cache sizes (fig 2) and competitive accuracy on the downstream tasks BOOLQ and ARC CHALLENGE (fig 3). However, there is typically a tradeoff between accuracy and latency and storage space. Therefore, the authors should also report the latency and storage space of the various tested methods.

The authors have only performed experiments with a version of Owen, and I would like to see whether their results generalize to another LLM, such as a Llama model.

While the learning approach of KVP is interesting and well-described, the main benefit of KVP remains unclear: is it better accuracy, reduced latency, reduced storage space, etc.?

**Questions:**

What is the latency and storage space of the various tested methods?

Are the perplexity gains reported in figure 2 statistically significant?

Why do you use different performance metrics for RULER (accuracy) and for OASST2 (perplexity)?

How does KVP work for another LLM, such as a Llama model?

What is the main benefit of KVP: better accuracy, reduced latency, reduced storage space, etc.?

---

> ### Author Response · Authors · 2025-11-26
>
> We thank the reviewer for their thorough assessment and valuable feedback. We are encouraged that the reviewer found our formulation of KV cache eviction as a learning problem to be original and novel, and our paper to be written clearly. We have carefully considered the weaknesses and questions raised, and we believe our revisions and new experiments address these points and strengthen the paper's contribution.
>
> **1. The Main Benefit of KVP: Adaptability and Generalization.**
> The primary benefit of KVP is its ability to learn an adaptive, model-agnostic eviction policy, in contrast to heuristic methods that rely on brittle, model-specific patterns. Hand-crafted strategies like StreamingLLM or KeyDiff are designed based on observed attention patterns in specific models. These patterns may not hold for different architectures or model sizes. To empirically validate this and address the generalization concern, we have added two significant new experiments. First, we evaluated KVP on a different and larger model, Microsoft's Phi-4 14B (Figure 6), to test architectural generalization. Second, we performed a zero-shot evaluation on the GovReport subtask from the LongBench benchmark (Figure 4) to test generalization to unseen tasks with very long contexts.  These experiments reveal several key findings:
> * Architectural Generalization: KVP's performance remains consistently strong on Phi-4 (Figure 6), while the relative performance of heuristic baselines changes significantly, highlighting their model-dependent nature and KVP's architectural robustness. Importantly, we used the same hyperparameters as in the Qwen2.5 experiments, further highlighting the strong generalizability of our approach across model families.
> * Long-Context & Out-of-Domain Generalization: KVP achieves the best performance on the GovReport task (Figure 4). This result is particularly strong as our agent, trained on RULER or OASST2, can zero-shot generalize not only to a new task but also to contexts significantly longer than those seen during training.
>
> Taken together, these new results make the main benefit of KVP clear: it provides a general and robust framework that learns to adapt to the specific patterns of any given model and task, rather than relying on fixed heuristics that fail to generalize across architectures or to extreme context lengths.
>
> **2. Efficiency: Latency and Storage Overhead.**
> We thank the reviewer for the important question regarding latency and storage overhead.
> - Storage Overhead: KVP's storage overhead is minimal. As detailed in Appendix A.3, our system consists of 112 small MLP agents (e.g., for the  Qwen model), each with ~650K parameters. The total checkpoint size for each agent  is just 2.6MB. This is negligible compared to the tens of gigabytes required for the base LLM.
> - Latency Overhead: To assess computational cost independent of hardware optimization (e.g., kernel fusion), we quantify the exact FLOPs overhead per token. :
>   1. Autoregressive Generation (Throughput). Zero Overhead. In our setting, the KV cache is compressed only once after prefill. This means KVP introduces no additional FLOPs during the decoding phase. Consequently, its throughput is identical to any other eviction method at the same cache budget. KVP's contribution is achieving superior accuracy for that given level of throughput.
>   2. Prefill Latency. Cost: 14.00 GFLOPs ->  14.15 GFLOPs. The overhead is strictly confined to the prefill stage. Based on the standard approximation of $2 * N_{params}$ FLOPs per token:
>       - Base model (Qwen-7B): 14.00 GFLOPs/token.
>       - KVP Overhead: Our 112 agents (0.65M params each) add 72.8M parameters, contributing an additional 0.15 GFLOPs/token.
>       - Total: The prefill cost becomes 14.15 GFLOPs/token, a marginal increase of ~1%.
>
>   This quantitative breakdown demonstrates that the computational footprint of KVP is negligible compared to the backbone model.
>
> [1/2]

---

> > ### Author Response · Authors · 2025-11-26
> >
> > **3. Clarifications on Experimental Protocol.**
> > Choice of Metrics (Accuracy vs. Perplexity): The use of different metrics for RULER and OASST2 is dictated by the benchmarks themselves.
> > *  RULER is a task-oriented benchmark with a definitive ground truth for each subtask. We use its official accuracy metric, as it directly measures success on the task.
> > * OASST2 is a dataset of open-ended, "in-the-wild" human-chatbot conversations, which lack a single correct "ground truth" response. For such generative scenarios, perplexity is the standard and most appropriate metric to evaluate how well the model's predictive distribution is preserved after compression.
> > * Statistical Significance of Perplexity Gains: The perplexity curves in Figure 2 are averaged over hundreds of samples from the test set. The consistent and smooth trend, where KVP outperforms baselines across a wide range of cache budgets, provides strong evidence of a systematic improvement and suggests the observed gains are not an artifact of noise.
> >
> > We believe these additions and clarifications, especially the new experiments on a larger model and longer context sizes, directly address the reviewer's concerns and make a much stronger case for the benefits of our learned approach. We are grateful for the opportunity to improve our work and hope the reviewer will find our revised manuscript more compelling.
> >
> > [2/2]

---

### Official Review · Reviewer_6mwy · 2025-11-01

**Soundness:** 2
**Presentation:** 3
**Contribution:** 2
**Rating:** 4
**Confidence:** 3

**Summary:**

The paper recasts KV-cache eviction as a reinforcement-learning (RL) ranking problem and proposes KV Policy (KVP). This is a lightweight policy that has per-head agents that learn to rank tokens by predicted future usefulness using only cache features, improving both efficiency and downstream accuracy. Under some low uniqueness and nestedness assumptions, eviction reduces to learning a single budget-agnostic ranking. KVP instantiates a Plackett–Luce stochastic ranking policy and leverages Gumbel-Sort for parallel, one-shot permutation sampling.  Each head’s agent is a small MLP scoring (key, value, position) and is trained offline on precomputed traces with a global reward equal to the negative cumulative future attention of evicted tokens summed over all budgets; the reward is normalized and optimized with REINFORCE using an RLOO baseline.  At inference time, the learned per-head rankings evict the lowest-ranked entries for any budget using only K/V/position (no queries or attention), requiring no extra LLM calls. The paper shows that KVP outperforms strong heuristics and attention-based baselines on RULER accuracy and OASST2-4k perplexity across most cache sizes. Additionaly, it generalizes competitively in zero-shot tests on BoolQ and ARC.

**Strengths:**

The paper provides a principled, budget-agnostic formulation of KV eviction and a practical, lightweight per-head policy that uses only cache-local features. Due to this, the technique is fast, query-free, and easy to deploy. Empirically, it outperforms strong heuristics and attention-aware baselines across cache sizes and tasks, showing robust generalization.

**Weaknesses:**

Training optimizes a proxy based on future attention computed from offline Q/K/V traces. This can misalign with downstream utility and requires precomputing and storing full-sequence Q/K/V (attention matrices omitted only due to size).

**Questions:**

1. Your reduction to a single budget-agnostic ranking hinges on uniqueness and nestedness. Do you have empirical evidence that generation traces satisfy these, and how sensitive is performance when nesting is violated (for instance, due to head/layer complementarity)?

2. Since KVP is attention-free and trained on offline "future-attention" signals, can you quantify alignment with downstream metrics vs. attention-aware policies that exploit query-specific information at prefil?

3. Could you add direct comparisons (same prefill-then-compress protocol and absolute-budget axes) to KeyFormer and MorphKV (Dialogue without Limits), and discuss where their key-centric selection/compression differs from your per-head RL ranking? Also, would their ideas change your conclusion about using a uniform per-head/layer budget?

---

> ### Author Response · Authors · 2025-11-26
>
> We sincerely thank the reviewer for their detailed summary and insightful feedback. We are encouraged that they recognize the principled formulation, practical lightweight design, and strong empirical performance of our work. The reviewer's questions have helped us clarify key aspects of our method and its evaluation.
> In response to the feedback, we have added two significant new experiments to further validate KVP's generalization capabilities: (1) a full evaluation on the Phi-4 14B model to demonstrate architectural generalization (Figure 6), and (2) a new zero-shot generalization experiment a subset of the LongBench benchmark to test performance on very long contexts and out-of-domain data (Figure 4). We believe these clarifications and new results address the reviewer's concerns and strengthen the paper.
>
> **Regarding the main weakness: Offline training on a proxy reward.**
> The reviewer raises a crucial point regarding the proxy reward. We view the use of this proxy not as a weakness or a trade-off, but as a fundamental strength of our design. By decoupling the utility estimation (trained offline on global future attention) from the inference process, we achieve a "best-of-both-worlds" scenario: we capture the rich, forward-looking information inherent in the attention mechanism without paying the latency cost of computing LLM inference online. Our design choice prioritizes creating a lightweight, query-free policy that can be trained once offline and deployed efficiently. The alternative, using online, query-specific information, introduces significant inference latency, which we aimed to avoid. Our strong empirical results suggest our proxy is highly effective, and this is further substantiated by our new zero-shot experiments on the GovReport task in the LongBench benchmark. In these experiments, shown in Figure 4, KVP trained only on RULER or on OASST2, generalizes effectively to this unseen, very-long-context task, providing strong evidence that our offline-trained policy captures a robust and generalizable utility signal. We further elaborate on the alignment in our answer to Q2.
>
> **Q1: On the uniqueness and nestedness assumptions.**
> We thank the reviewer for this sharp question about the theoretical underpinnings of our budget-agnostic approach. The uniqueness and nestedness assumptions are indeed theoretical idealizations that allow us to reduce an intractable problem (learning a separate eviction policy for every possible budget) to a much simpler one: learning a single, universal ranking of tokens. We do not expect these properties to hold perfectly in practice, especially given complex phenomena like head/layer complementarity. The most compelling empirical evidence for the validity of this simplification is the core result of our paper: KVP's strong performance across the entire range of cache budgets. If violations of the nestedness assumption were causing significant performance degradation, we would expect to see our single-ranking policy fail dramatically at certain budget sizes. Instead, KVP consistently outperforms baselines across different compression ratios. This demonstrates that while minor violations may occur, they are not catastrophic, and a single learned ranking serves as a robust and highly effective policy in practice.
>
> **Q2: Quantifying alignment with downstream metrics vs. attention-aware policies.**
> This is an excellent question. While a direct mathematical function mapping our proxy reward to a downstream metric like PPL or accuracy is challenging to derive, we can provide strong theoretical and empirical arguments for its alignment.
> - Theoretical Alignment: By construction, our proxy is perfectly aligned in the ideal case. If a set of tokens will receive a cumulative future attention score of zero from all subsequent queries, evicting them is mathematically guaranteed to produce an output identical to that of a full-sized cache. This is because those tokens would have a weight of zero in all future attention computations. Our RL agent is trained to approximate this exact future utility.
> - Empirical Alignment: Our experiments provide strong evidence of this alignment. KVP performs well on downstream tasks, this indicates that the global, forward-looking utility captured by our offline-trained policy is a reasonable signal. Furthermore, our new experiments confirm this trend across two dimensions: (1) our evaluation on Phi-4 14B (Figure 6) shows generalization across architectures, and (2) our new zero-shot evaluation on the LongBench GovReport (Figure 4) shows that KVP excels on a downstream task with contexts significantly longer and from a different domain than its training data. This directly demonstrates the alignment of our proxy with performance on challenging, out-of-distribution tasks.
>
> [1/2]

---

> > ### Author Response · Authors · 2025-11-26
> >
> > **Q3: Comparison to KeyFormer and MorphKV.**
> > We thank the reviewer for suggesting these important related works and for the opportunity to clarify our experimental focus. While all three methods aim to reduce the KV cache, they operate on fundamentally different principles, making a direct experimental comparison complex.
> > - Experimental Scope and Baseline Choices: The primary focus of our evaluation is on attention-free eviction policies. These methods are highly efficient as they operate using only cache-local features, avoiding any query-dependent computations. Our core comparison is therefore against strong attention-free heuristics like KeyDiff. We included attention-aware policies, e.g., SnapKV and TOVA primarily as a cross-category reference point, and found ourselves surprisingly competitive against them. This finding validates our main claim: a well-trained, query-free policy can be effective, making an additional comparison with KeyFormer (which selects based on global attention aggregation rather than learned local utility) less critical for establishing this point. We have clarified this distinction in our revised Related Work.
> > - Conceptual Differences: MorphKV performs state compression by identifying and merging semantically similar KV pairs into new, synthetic representations. We view MorphKV and KVP as orthogonal and complementary: KVP focuses on eviction ranking, identifying which tokens are more or less useful. These approaches operate on different axes: one could utilize KVP to filter out the least useful tokens and then apply MorphKV’s merging logic to the remaining high-utility tokens. We have clarified these distinctions in our revised Related Work.
> > - Non-uniform budgeting: The reviewer also raises an excellent point regarding the uniform per-head budget. For this specific study, we strictly adhered to a uniform budget to ensure a fair, controlled comparison focused purely on the quality of the eviction strategy itself, rather than on budget allocation heuristics. A non-uniform budget is a promising direction. In fact, our per-head rankings from KVP could also directly inform such a dynamic allocation scheme (e.g., allocating more budget to heads that assign higher utility scores to more tokens). We have noted this as a key direction for future work.
> >
> > We believe these clarifications and our new experimental results on Phi-4 14B (Figure 6) and LongBench (Figure 4) significantly strengthen the paper. We hope our responses have adequately addressed the reviewer's concerns.
> >
> > [2/2]

---

> > > ### Comment · Reviewer_6mwy · 2025-11-27
> > >
> > > Thank you for the reply. While I agree with your statements, I think a more quantitative comparison with some of these works would have been insightful. It would have helped understand if these seemingly orthogonal techniques do/do not influence each other. Nevertheless, the paper is a good step in the right direction, I remain optimistic.

---

> > > > ### Author Response · Authors · 2025-12-01
> > > >
> > > > We sincerely thank the reviewer for their timely engagement and optimistic assessment. We agree that a quantitative exploration of the synergies with orthogonal methods is an insightful direction for future work, and we are grateful for their constructive feedback which has helped strengthen the paper.

---

### Official Review · Reviewer_TTjf · 2025-11-02

**Soundness:** 3
**Presentation:** 3
**Contribution:** 3
**Rating:** 6
**Confidence:** 4

**Summary:**

In this paper, the authors propose KV Policy (KVP), a reinforcement learning (RL) framework that reframes cache eviction as a learning-to-rank problem. Each attention head in the transformer is assigned a lightweight RL agent trained on precomputed generation traces to predict each token's future utility using only key, value, and positional embeddings. Empirical results on RULER and OASST2-4k show that KVP outperforms other attention-based (e.g., SnapKV, TOVA) and attention-free (e.g., StreamingLLM, KeyDiff) baselines in accuracy and perplexity under the same budgets.

**Strengths:**

S1. This paper tackles an important problem of KV cache compression by eviction.

S2. This paper proposes to learn to evict the KV states, which is less explore in the area.

S3. The paper is well written and structured.

**Weaknesses:**

W1. Comparisons with recent sparse kv cache retrieval approaches, e.g., IceCache, ArkVale, MagicPig, InfiniGen, should also be included in addition to the kv cache eviction approaches.

W2. More backbone LLMs should be included. Qwen2.5 should be upgraded to Qwen3-8B. At least Llama3.1-8B should be included for the different variety of the models. One of a medium size LLMs, i.e., ~32B, should also be included to demonstrate the scalability of the proposed approach.

W3. More long-context benchmarks, e.g., longbench, should be included. Moreover, long-generation benchmarks, e.g., longgenbench, should also be included in experiments.

**Questions:**

Q1. What is the training cost, i.e., training data size, training time, etc., of the RL-based approach?

Q2. Would the RL-based approach be able to be combined with the heuristics-based approaches?

Q3. Can you show some failure case studies, such that we can better understand the pros and cons of the RL-based approach?

---

> ### Author Response · Authors · 2025-11-26
>
> We sincerely thank the reviewer for their thoughtful feedback and constructive suggestions. We are encouraged that the reviewer found our work well-written, our RL-based approach novel, and the problem of KV cache eviction important.
>
> In response to the feedback, we have conducted new experiments to test the scalability of our approach, added further analysis on training costs and failure cases, and clarified the scope of our comparisons. We believe these additions significantly strengthen the paper. In particular, we have added two significant new experiments: (1) a full evaluation on the Phi-4 14B model to demonstrate scalability (Figure 6), and (2) a new zero-shot generalization experiment on a subset of the LongBench benchmark to test performance on very long contexts (Figure 5).
>
> Below, we address the reviewer's specific points.
>
> **W1: Comparisons with recent sparse KV cache retrieval approaches.**
> We thank the reviewer for raising this important point and for suggesting these relevant works. We agree that methods managing memory hierarchies are a crucial part of the landscape for efficient inference. We see these approaches not as direct competitors, but as orthogonal and complementary to our work. Methods like ArkVale manage a memory hierarchy, offloading KV entries to slower CPU memory. A critical component of such systems is the policy that decides which tokens to offload. This is precisely where KVP can provide a significant benefit. Our learned KVP agents produce a principled, forward-looking ranking of each token's future utility. This ranking provides an ideal, data-driven signal for an offloading policy: the lowest-ranked tokens are the natural candidates to be moved to slower memory. Using KVP to guide the offloading decisions could make these hierarchical systems more efficient by minimizing costly reloads, representing a powerful synergy between our approaches.  For the experimental section of our paper, we deliberately focused on a direct comparison with other pure eviction methods. This was to ensure a fair, apples-to-apples evaluation of the core ranking policy's quality, isolating it from the different system-level trade-offs (e.g., reload latency) introduced by offloading. We have updated our related work section (Section 2) to clarify this distinction and to explicitly discuss the exciting potential for combining KVP with memory hierarchy management techniques as a promising direction for future research.
>
> **W2: More backbone LLMs should be included.**
> We agree that demonstrating the scalability and generalizability of KVP across different model architectures and sizes is crucial. To this end, we have run a full set of new experiments on a larger and architecturally different model: Microsoft's Phi-4 14B. Importantly, we used the same hyperparameters as in the Qwen2.5 experiments, further highlighting the strong generalizability and robustness of our approach across model families. Our results, in Figure 6, show that KVP continues to outperform all baselines, demonstrating its effectiveness is not limited to the Qwen architecture. Interestingly, we observed that the relative performance of the heuristic baselines changes between models, highlighting the benefit of our learned, adaptive policy which can specialize to the unique attention patterns of each LLM. We have added these results to the appendix. While we could not secure a ~32B model for the rebuttal period due to computational constraints, we agree this is an important next step and have noted it as a key direction for future work.
>
> **W3: More long-context benchmarks should be included.**
> This is an excellent suggestion. To directly address this point, we conducted a new zero-shot generalization experiment on the "GovReport" task of LongBench. Importantly, this subtask was selected at random from the LongBench long-context tasks as a full evaluation on the entire benchmark was not feasible during the rebuttal period. We evaluated both our RULER-trained and OASST2-trained KVP agents on this unseen, long-context task. The results, presented in Figure 4, show that KVP  outperforms all baselines. This new finding is powerful because it simultaneously demonstrates that our method (1) generalizes effectively to context lengths far exceeding its training data, and (2) provides another strong confirmation that the learned policy generalizes zero-shot to entirely new tasks and data distributions. We believe this targeted experiment strongly validates KVP's robustness and generalizability.
>
> [1/2]

---

> > ### Author Response · Authors · 2025-11-26
> >
> > **Q1: What is the training cost of the RL-based approach?**
> > The training for KVP is designed to be highly efficient and lightweight.
> > * Data Generation: This is the primary one-time cost, requiring a single forward pass over the training samples to collect and store on disk the K, V and Q embeddings for all KV caches of the entire generation. This is approximately equivalent to the cost of standard inference once on the dataset.
> > * Training Data: The agents are trained on pre-computed traces. For RULER, we used ~6,000 samples, and for OASST2, ~4,500 samples.
> > * Model Size: Each per-head agent is a small MLP with only ~650K parameters, resulting in a checkpoint size of just 2.6MB.
> > * Training Time: The agent training completes in less than 30 minutes on 8 NVIDIA H100 GPUs. The code is not yet fully optimized, and hyperparameters not tuned for training speed, suggesting further speed-ups are possible.
> > This demonstrates that our offline training framework has a very low computational footprint, making it a practical and scalable solution. We have added these details to Appendix A.3.
> >
> > **Q2: Would the RL-based approach be able to be combined with the heuristics-based approaches?**
> > Yes, absolutely. KVP is not only compatible with heuristics but is already combined with one in our implementation. As described in Appendix A.2, our policy always retains the first 4 and last 16 tokens (i.e., StreamingLLM), applying the learned RL ranking only to the tokens in between. This hybrid approach leverages the strengths of both paradigms. We agree that exploring more advanced combinations, not limited to the eviction, is a very promising direction. For instance, instead of a uniform budget, KVP's rankings could inform a dynamic budgeting scheduler that allocates more memory to more important heads.
> >
> > **Q3: Can you show some failure case studies?**
> > Thank you for this insightful question. We found that identifying clear, interpretable "failures" in downstream tasks is difficult, as performance degradation often results from many small, distributed ranking errors rather than a single catastrophic one. To provide more insight, we have added a qualitative analysis in the appendix in Figure 8. We include a visualization (similar to Figure 1) where we compare the rankings produced by KVP and other baselines against the ground-truth future attention scores. Our analysis in Figure 8 shows that even in this challenging case, KVP's ranking more closely approximates the global importance distribution of the oracle than competing heuristics do.
> >
> > [2/2]

---

### Author Response · Authors · 2025-11-26
**Summary of Major Revisions**

We sincerely thank all reviewers for their thoughtful feedback. In response, we have substantially strengthened the paper by directly addressing the primary concerns regarding generalization and scalability with two major new experiments:
* **Zero-shot generalization to long contexts (new evaluation on LongBench, in Figure 4)**: We added a zero-shot evaluation on LongBench. KVP significantly outperforms all baselines, demonstrating its ability to generalize to contexts lengths and data distributions far beyond its training set.
* **Scalability to other and larger models (new evaluation on Phi-4 14B, in Figure 6)**: We conducted a full evaluation on Phi-4 14B. KVP's strong performance confirms its effectiveness is not limited to a single architecture and that our learning framework adapts successfully to different models.
* **Latency, Training Cost, and Related Work**: We have incorporated detailed analyses of KVP's negligible latency overhead and low training cost in Appendix A.3, and have clarified the positioning of our work relative to complementary methods in Section 2.

We believe these new results and clarifications directly address the reviewers' primary concerns and make a much stronger case for the benefits of our learned approach. We are grateful for the opportunity to improve our work and hope the reviewers will agree.

---

### Author Response · Authors · 2025-12-01
**Summary for the New Area Chair**

Dear new Area Chair,

In light of the unusual circumstances, we have prepared this brief summary of the review discussion and our rebuttal.

 First, we were encouraged that all four reviewers reached a consensus on the work core strengths. The RL formulation was praised as "original" (`xyfd`), "principled" (`6mwy`) and “less explored” (`TTjf`). Crucially, its lightweight, query-free design was highlighted as a key practical strength, enabling "no overhead at inference time" (`6mwy`, `xyfd`). The paper was commended as "well written" (`TTjf`, `xyfd`), and even our most critical reviewer `DJnn` highlighted its scientific rigor, praising the "careful ablations" (`DJnn`, `xyfd`) and "well-motivated design” (`DJnn`).

The primary concerns centered on generalization, scalability, and efficiency. We took this feedback seriously and performed substantial new experiments to address every point:

- **Concern**: Generalization to other models and "truly long" contexts. (Raised by `TTjf`, `xyfd`, `DJnn`)
    - **Our Action**: We ran two major new experiments:
        - A full evaluation on a larger, different model family, **Phi-4 14B** (Fig. 6). KVP’s performance remained strong, comparable to the model consider initially (Qwen 2.5 7B), while the relative effectiveness of baselines changed; demonstrating KVP’s superior robustness across architectures.
        - A zero-shot evaluation on a **LongBench subtask** (Fig. 4), showing KVP excels on unseen tasks and contexts far longer than its training data, and it is considerably better than baselines.
- **Concern**: Unclear latency, training, and storage costs. (Raised by `xyfd`, `DJnn`, `TTjf`)
    - **Our Action**: We added a detailed analysis (Appx. A.3) quantifying KVP's negligible overhead: zero during generation, a ~1% FLOPs increase at prefill, agent’s checkpoint of 2.6MB, and efficient training (<30 mins on 8 GPUs).
* **Concern**: Comparisons with orthogonal/concurrent work. (Raised by `TTjf`, `6mwy`, `DJnn`)
    * **Our Action**: We clarified in Sec. 2 how KVP is a pure eviction policy, making it complementary to, not a direct competitor of, memory-hierarchy management (e.g., ArkVale) or state-compression (e.g., MorphKV) methods.

**Rebuttal Timeline and Interrupted Discussion**
The scale of these experiments meant our rebuttal was posted on Nov 26th. We saw immediate positive traction. Within a day, one reviewer raised their score from 4 to 6, commenting, *“the paper is a good step in the right direction, I remain optimistic.”*

Unfortunately, this score was administratively reset and the discussion period was halted on Nov 28th, preventing other reviewers from weighing in on our substantial revisions. Given our new experiments directly addressed the primary concerns of all reviewers, we are confident the discussion would have continued positively.

Thank you for your careful consideration. We believe the revised paper makes a much stronger case for acceptance and hope this summary is helpful.

Sincerely,

The Authors

---

### Meta-Review · Area_Chair_WV77 · 2026-01-06

**Summary:**

This paper proposes KV Policy (KVP), framing KV-cache eviction as a per-head RL learning-to-rank problem using only (K,V,position) features, targeting a query-free eviction policy (no query/attention dependence at inference). Reviewers broadly agree the formulation is original and practically motivated (e.g., “no overhead at inference time” framing), and the paper is clearly written. However, the decision hinges on several persistent concerns: (i) limited/unclear comparisons to relevant learned or system-level approaches, (ii) incomplete evidence on end-to-end efficiency (especially wall-clock latency/throughput), and (iii) questions about robustness/generalization and the proxy objective alignment.

**Reviewer Concerns:**

Concerns largely addressed by the rebuttal / discussion:

+Generalization beyond the original backbone and to longer contexts: authors added experiments on Phi-4 14B and a LongBench (GovReport) zero-shot evaluation to support cross-model and longer-context generalization.

+Training cost transparency: authors provided concrete training details (data sizes, <30 min on 8×H100, small agent checkpoints) and argued overhead is negligible.

+Clarification of scope vs. “orthogonal” methods (e.g., offloading/compression systems): authors clarify KVP as a pure eviction/ranking component and argue complementarity.

Concerns still outstanding:

-Efficiency evidence remains indirect: key reviewers asked for wall-clock latency/throughput or speedups; the rebuttal mostly substitutes a FLOPs-based estimate and “zero overhead during decoding” under their protocol, but does not provide the runtime comparisons requested.

-Comparative evaluation gaps: multiple reviewers requested broader comparisons (e.g., additional long-context benchmarks; and/or direct comparisons to other learned KV reduction methods). While the rebuttal discusses why some are “not direct competitors,” the lack of direct quantitative comparisons leaves the empirical case less convincing.

-Methodological concerns about the proxy objective and assumptions: concerns remain that optimizing a future-attention proxy and relying on theoretical assumptions (nestedness/uniqueness) may not faithfully track downstream utility across settings.

**Reviewer Scores:**

Given the additional experiments/clarifications already posted, I would not expect significant changes of review score.

---

### Decision · Program_Chairs · 2026-01-26

Reject